# High-Dimensional Prediction for Sequential Decision Making

**Georgy Noarov** [1]   **Ramya Ramalingam** [1]   **Aaron Roth** [1]   **Stephan Xie** [2]

## Abstract

We give an efficient algorithm for producing multi-dimensional forecasts in an online adversarial environment that have low bias subject to any polynomial number of conditioning events, that can depend both on external context and on our predictions themselves. We demonstrate the use of this algorithm with several applications. We show how to make predictions that can be transparently consumed by any polynomial number of downstream decision makers with different utility functions, guaranteeing them diminishing swap regret at optimal rates. We also give the first efficient algorithms for guaranteeing diminishing conditional regret in online combinatorial optimization problems for an arbitrary polynomial number of conditioning events — i.e. on an arbitrary number of intersecting subsequences determined both by context and our own predictions. Finally, we give the first efficient algorithm for online multicalibration with $O(T^{2/3})$ rates in the ECE metric.

## 1. Introduction

Decision making in sequential settings is a challenging problem from both the theoretical and the applied perspective. Tackling the nonstationarity inherent to sequential interactive decision making settings is the subject of the online learning, reinforcement learning, and bandits literatures (see e.g. Slivkins et al. (2019)).

The dominant paradigm is *regret minimization*, which places emphasis on optimizing various notions of *regret* for decision-making agents, thus ensuring that these agents' cumulative rewards stay ahead of nontrivial benchmark classes of strategies. The most basic and tractable notion of regret, called *external* regret (with the qualifier "external" usually

omitted), compares the utility of an agent's play to that of the best-in-hindsight fixed strategy. It is a *marginal* guarantee, in the sense that it forces the agent to do as well as the benchmark cumulatively over all rounds $t = 1 \ldots T$, but does not imply locally optimal performance on any subset of rounds. Thus, various extensions to more challenging benchmark classes have been developed, giving rise to notions such as swap regret, adaptive regret, and beyond.

Typically, regret minimization algorithms directly optimize the agent's loss or reward function. This works well in single-agent settings and for simple regret guarantees. However, this direct approach does not naturally address sequential environments which (a) involve multiple decision-making agents with different utilities/rewards; or (b) demand performance guarantees *conditionally* on the agents' own actions. Motivated by these challenges, in this paper we take a different route and shift from direct reward optimization to a more subtle but flexible approach.

---

**Recipe**

1. Distill the decision-making-relevant aspects of the sequential environment into a sequence of "sufficient statistics": vector-valued *states* of the environment, such that all agents could easily identify their reward-maximizing actions *if* they perfectly knew the next state.

2. Devise an efficient online adversarial algorithm that predicts the upcoming state of the environment with such granular accuracy guarantees that the agents can *treat our predictions as the true states* for the purposes of picking reward-maximizing actions.

---

Step 1 of this recipe will naturally always require insight into the specific setting at hand, and identifying appropriate sufficient statistics can often be an art. However — as we will show in this paper — Step 2, which may appear just as hard to execute, turns out to have a "silver-bullet" solution that applies to a wide variety of complex decision-making settings: our *event-unbiased prediction* framework.

### 1.1. Our Contributions

Our central contribution is a general and efficient algorithmic framework for *event-unbiased prediction* of vector quan-

---

[1]Department of Computer and Information Science, University of Pennsylvania [2]Machine Learning Department, Carnegie Mellon University. Correspondence to: Georgy Noarov <gnoarov@seas.upenn.edu>.

*Proceedings of the 42nd International Conference on Machine Learning*, Vancouver, Canada. PMLR 267, 2025. Copyright 2025 by the author(s).

tities in the online adversarial setting. It accommodates arbitrary finite collections of (unweighted or weighted) *conditioning events*, which may depend on external contexts or on the predictions themselves, and guarantees optimally converging bias conditional on each of these events. We introduce the framework, and formally derive the algorithm and its guarantees, in Section 2.

We then show how our framework implements the above recipe in complex environments shared by one or more decision-making agents. In this context, as our Recipe prescribes, we use it in a "Predict-then-Act" manner, which allows us to bypass direct optimization of policy rewards for any given agent, instead focusing on forecasting the evolving *state* of the environment, which the agents' rewards ultimately depend on. The main advantages this offers are:

(1) Strong *conditional* regret guarantees: By leveraging our predictions, the agents can minimize their regret not just marginally over the whole sequence of the interaction, but also on arbitrary context- and action- defined subsequences of the rounds. This is enabled by our framework's ability to de-bias its state predictions over arbitrary such subsequences of rounds, thus endowing the agents' decisions with corresponding regret guarantees on these subsequences.

(2) *Coordination* of multiple agents: Instead of letting all agents take the burden of running their respective regret minimization algorithms locally, we are able to instead provide a single public-knowledge state prediction, which all agents can then use to inform their actions in a straightforward way: They can just *best-respond* to the announced state.

We illustrate the power of our framework on the following concrete examples:

**Online Combinatorial Optimization** We consider a general *online combinatorial optimization* setting, which models a sequential interaction of multiple agents with combinatorially large action spaces corresponding to structured subsets of some set of $d$ base actions, and which encompasses a variety of classical problems such as online routing. We show how to endow all agents with conditional regret guarantees, letting the agents incur only sublinear regret on any given finite collection of subsequences defined by external contexts and even by their *own actions*.

An example instantiation is the online routing problem in which multiple agents are trying to get from home to work as fast as possible every day. We can issue *trustworthy* daily road congestion forecasts, such that each agent who always selects the fastest route according to our forecast is guaranteed no regret conditional both on salient covariate information (such as the weather) and on their own choice of route (e.g. downtown route; interstate route; route visiting their favorite coffee shop). Thus, each agent will then be happy with their average travel time not just overall, but

also, e.g., over those days when it rained and they went to get coffee on their way to work; or over those days when their local football team was playing.

Our algorithm has an efficient dependence on the dimension $d$ of the combinatorial problem (e.g. $d$ is the number of edges in online routing): it runs using only $poly(d)$ calls to the offline oracle for the problem (e.g. a shortest-path algorithm for routing) and has $\widetilde{O}(d)$ regret dependence. Prior techniques for giving conditional regret guarantees by direct regret minimization (e.g. the algorithm of Blum & Mansour (2007)) have running time scaling with the number of actions (e.g. paths in online routing), which can be exponential in $d$ in online combinatorial optimization. See Section 4.

**Swap Regret for Multiple Agents** In the classical experts setting but with multiple regret-minimizing agents, we show how to use our framework to issue a single coordinating prediction at all rounds that will guarantee *swap regret* at optimal rates to all agents simultaneously. See Section 3.

**Online Multicalibration** In Section 2.2 we show that our algorithm, when appropriately instantiated, gives the first efficient $O(T^{2/3})$ online multicalibration algorithm. Multicalibration (Hébert-Johnson et al., 2018) is a strengthening of the classical statistical concept of calibration (Dawid, 1985), and promises calibrated predictions on rich collections of contextually defined groups in the data. The best-known efficient algorithm for (vanilla) online calibration is due to Abernethy et al. (2011), and we match its rate in the more challenging multicalibration setting. The best-known multicalibration bound achieved in the ECE metric was $O(T^{3/4})$ (Gupta et al., 2022).

## 1.2. Follow-up Work and Impact

Since the appearance of the preprint of this paper, our methodology has been employed in several subsequent works, showing the broad utility of our framework.

Collina et al. (2024b) apply our algorithms in a repeated principal-agent setting defined by Camara et al. (2020) to obtain exponentially improved bounds. Briefly, Camara et al. (2020) gave a mechanism that replaced the standard "common prior" assumptions that underlie principal-agent models with calibrated forecasts of an underlying state, and is applicable in adversarial settings. Camara et al. (2020) use the traditional notion of calibration, and as a result inherit exponential computational and statistical dependencies on the cardinality of the state space. Collina et al. (2024b) show how to apply our techniques to recover the same results (under weaker assumptions) with an exponentially improved dependence on the cardinality of the state space.

Roth & Shi (2024) apply our algorithms to produce forecasts that guarantee *all* downstream decision makers $O(\sqrt{T})$ *swap regret* without the need to know their utilities in ad-

vance, improving on the simultaneous no-external regret guarantees of Kleinberg et al. (2023).

Hu & Wu (2024), using our algorithm, further remove any dependence on the cardinality of the decision maker's action space in the 1-dimensional setting.

Collina et al. (2024a) use our algorithm as part of their construction giving computationally tractable "agreement" protocols generalizing Aumann's agreement theorem.

Our original extended preprint contains another application of our algorithm, to decision conditional "score-free" conformal prediction. By appropriately de-biasing the scores of any online multiclass predictor, we can make them look like correct class probability vectors to downstream prediction set algorithms, letting prediction sets with valid conditional coverage guarantees be simply "read off" from the multiclass probability vectors.

### 1.3. Related Work

Our framework extends an array of recent prior works on decision-focused prediction. Zhao et al. (2021) introduce decision calibration: a calibration-based framework for the offline (batch) setting. Decision calibration is less expressive than our event-unbiasedness notion: e.g., unlike our framework, it does not imply swap regret guarantees. Gopalan et al. (2022a) introduce omniprediction: an approach for making predictions that simultaneously can be used to optimize multiple downstream loss functions. Omniprediction is related to our Predict-then-Act approach to optimizing utilities for multiple agents: the omni-predictions need to be post-processed downstream to optimize each loss, but can be treated as if they are real probabilities in that downstream optimization. Dwork et al. (2021) study outcome indistinguishability: a complexity-theoretic perspective on making predictions that appear indistinguishable from the ground truth to a rich class of distinguishers. These works have generated considerable follow-up research, which we discuss in Appendix A. We extend these insights into a broadly applicable, efficient *online adversarial* framework with optimal-rate guarantees.

**No-Regret Guarantees in Online Learning** No-regret learning has been studied at least since Hannan (1957); see Hazan (2016) for a modern treatment of this literature. Kalai & Vempala (2005) gave efficient no regret algorithms in online linear and combinatorial optimization problems. Internal regret, which corresponds to regret on the subsequences defined by the play of each action, was first defined by Foster & Vohra (1999), who also showed it could be obtained by best responding to calibrated forecasts.

The seminal contribution of Foster & Vohra (1999) has led to a long list of works exploring the interplay of no-regret and online calibration algorithms, discussed in more detail in the Appendix. As one important precursor to our work, Haghtalab et al. (2023a) develop a general online multiobjective learning framework based on a game between no-regret and best-response algorithms, with the focus on deriving improved multicalibration guarantees in the online and batch settings. Their reduction to no-regret learning allows them, in particular, to obtain small-loss group-calibration bounds, mirroring our small-loss event-conditional bias bounds.

Lehrer (2003) defined a notion of "wide-range regret" which is equivalent to conditional regret: that a player should have no regret not just overall on the whole sequence of rounds, but also conditional on various events — subsequences that can be defined both as a function of time ("time selection functions") and as a function of the actions of the learner. Blum & Mansour (2007) gave algorithms for obtaining this kind of conditional regret guarantees (including, notably internal (or "swap") regret as a special case). The algorithm of Blum & Mansour (2007) is efficient when the action space is polynomially sized: it requires computing eigenvectors of a square matrix of dimension equal to the number of actions in the game. Motivated by fairness concerns, Blum & Lykouris (2020) give an algorithm for obtaining diminishing "groupwise" regret, which is equivalent to regret with respect to a collection of time selection functions. These results do not accommodate events that can depend on the actions of the learner, which are crucial for our applications.

## 2. General Framework and Algorithm

**Unbiased Prediction Setting**  Let $\mathcal{X}$ be a *context space* which can be arbitrary. Let the *state space* $\mathcal{S}$ be any convex and compact subset of $\mathbb{R}^d$ and assume without loss of generality that $\max_{s \in \mathcal{S}} \|s\|_\infty \leq 1$. Any element $s \in \mathcal{S}$ is called a *state*. The space of distributions over $\mathcal{S}$ is denoted $\Delta S$.

In this section, we consider the task of online adversarial contextual prediction of the states over $t \in [T] := \{1, \ldots, T\}$ time steps. The learner sequentially observes contexts $(x_t)_{t \in [T]} \in (\mathcal{X})^T$, and makes randomized state predictions $(\bar{s}_t)_{t \in [T]} \in (\Delta S)^T$. The *adversary* sequentially responds by generating the true states $(s_t)_{t \in [T]} \in (\mathcal{S})^T$.

The learner aims to make predictions *unbiased* conditional on a given collection $\mathcal{E} = (E_j)_{j \in [n]}$ of $n \geq 1$ *events*.

**Definition 2.1** (Event; Event-Conditional Bias). An *event* is a mapping $E : \mathcal{X} \times \mathcal{S} \to [0, 1]$; the event's value in round $t$ is $E(x_t, \hat{s}_t)$. If the range of $E$ is $\{0, 1\}$ then we call $E$ a *binary* event.
The cumulative *E-conditional bias* in coordinate $i \in [d]$ of the state predictions after $T$ rounds is defined as:

$$\text{Bias}_T(E, i) := \mathbb{E}_{\hat{s}_t \sim \bar{s}_t \forall t} \left[ \left| \sum_{t=1}^T E(x_t, \hat{s}_t) \cdot (\hat{s}_{t,i} - s_{t,i}) \right| \right].$$

The general protocol is as follows. In rounds $t = 1 \ldots T$:

1. The learner observes context $x_t \in \mathcal{X}$, and receives event functions $E(x_t, \cdot) : \mathcal{S} \to [0,1]$ for $E \in \mathcal{E}$.

2. The learner makes *state prediction* $\bar{s}^t \in \Delta\mathcal{S}$.

3. The adversary sees $\bar{s}^t$ and generates *true state* $s^t \in \mathcal{S}$.

4. The *(realized) prediction* $\hat{s}^t \in \mathcal{S}$ is sampled: $\hat{s}^t \sim \bar{s}^t$.

**Objective:** The learner's goal is to make predictions that are *unbiased* in all coordinates $i \in [d]$ conditional on all events $E \in \mathcal{E}$. In fact, we define our desideratum by requiring the bias rate conditional on every event to diminish as a function of the event's frequency (rather than as a function of the time horizon $T$); such strengthened bounds are referred to as *small-loss* in online learning.

**Definition 2.2** (Unbiased Prediction). Let $n_T(E) = \mathbb{E}_{\hat{s}_t \sim \bar{s}_t \forall t} \left[ \sum_{t=1}^{T} \left( E\left(x_t, \hat{s}_t\right) \right)^2 \right]$ denote the *incidence*[1] of event $E$. Then we call the learner's predictions *unbiased conditional on event collection $\mathcal{E}$* if for all events $E \in \mathcal{E}$,

$$\max_{i \in [d]} \mathrm{Bias}_T(E, i) = O\left( \log(d|\mathcal{E}|T) + \sqrt{n_T(E) \log(d|\mathcal{E}|T)} \right).$$

We further denote $\mathrm{Bias}_T(E) := \max_{i \in [d]} \mathrm{Bias}_T(E, i)$.

### 2.1. General Algorithm with Bounds

**OLO Primitives** We now develop a general algorithm that achieves the above bias bounds for any given finite event collection. It will rely on online linear optimization (OLO) methods. We briefly review the OLO protocol over any $d'$-dimensional convex domain $\mathcal{C} \subseteq \mathbb{R}^{d'}$. In rounds $t = 1 \ldots T$, an OLO algorithm $\mathcal{A}_{\mathrm{OLO}}$ plays some $c_t \in \mathcal{C}$, the adversary observes that and generates a loss vector $\ell_t \in \mathbb{R}^d$, and $\mathcal{A}_{\mathrm{OLO}}$ observes $\ell_t$ and suffers loss $\langle \ell_t, c_t \rangle$ in that round. The overall performance of $\mathcal{A}_{\mathrm{OLO}}$ is measured via *OLO regret* to the best point in $\mathcal{C}$ that could have been played. Letting the regret to any admissible point be defined as $\mathrm{Reg}_T(\mathcal{A}_{\mathrm{OLO}}, c) := \sum_{t=1}^{T} \langle \ell_t, c_t - c \rangle$, the OLO regret of $\mathcal{A}_{\mathrm{OLO}}$ is defined as:

$$\mathrm{Reg}_T(\mathcal{A}_{\mathrm{OLO}}) := \max_{c \in \mathcal{C}} \mathrm{Reg}_T(\mathcal{A}_{\mathrm{OLO}}, c),$$

Many OLO algorithms $\mathcal{A}_{\mathrm{OLO}}$ achieve the classic minimax regret bound $\mathrm{Reg}_T(\mathcal{A}_{\mathrm{OLO}}) = O(\sqrt{T})$ for all convex compact domains $\mathcal{C}$ and bounded losses; the simplest one is online gradient descent (OGD) of Zinkevich (2003).

However, for particular domains $\mathcal{C}$, algorithms with even stronger regret bounds have been developed. We will use

---

[1]Note: $n_T(E)$ is at most the expected count of $E$'s occurrences (i.e., rounds where $E(x_t, \hat{s}_t) = 1$), with equality for binary $E$.

one such method called MsMwC (Multiscale Multiplicative Weights with Correction) due to Chen et al. (2021) whose domain is the $d'$-dimensional simplex: $\mathcal{C} = \Delta_{d'}$. This special setting is also called the *experts setting*, as each of the vertices $(e_i)_{i \in [d']}$ of the simplex ($e_i \in \mathbb{R}^{d'}$ denoting the $i$th standard basis vector) can be viewed as an expert. Rather than only promising $O(\sqrt{T})$ regret to the best expert, MsMwC obtains small-loss bounds simultaneously to each expert, which scale with the losses of the expert:

**Theorem 2.3** (Theorem 2 of Chen et al. (2021)). *There exists an experts OLO algorithm $\mathcal{A}_{\mathrm{MsMwC}}$ with per-round time complexity $poly(d')$ for $d'$ experts, whose chosen points $w_t \in \Delta'_d, t \in [T]$, achieve the following regret bound to every expert $e_i \in [d']$ provided all losses $\ell_t \in [-1,1]^{d'}$:*

$$\mathrm{Reg}_T(\mathcal{A}_{\mathrm{MsMwC}}, e_i) = O\left( \log(d'T) + \sqrt{\log(d'T) \cdot \sum_{t=1}^{T} \ell_{t,i}^2} \right)$$

**The General Unbiased Prediction Algorithm** We now apply the OLO tools described above to obtain an efficient unbiased prediction algorithm that achieves the guarantee of Definition 2.2 for any finite event collection $\mathcal{E}$, state space $\mathcal{S}$ and feature space $\mathcal{X}$.

For notational convenience, we will represent any event collection $\mathcal{E} = (E_j)_{j \in [n]}$ as a single vector-valued *event function* $\vec{E} : \mathcal{X} \times \mathcal{S} \to [0,1]^n$. Definition 2.2 essentially requires us to learn to make randomized state predictions $(\hat{s}_t)_{t \in [T]}$ to optimize the quantity: $\Psi\left( (\hat{s}_t)_1^T, (s_t)_1^T \right) := \max_{i \in [d], j \in [n]} \left| \sum_{t=1}^{T} \vec{E}_j(x_t, \hat{s}_t) \cdot (\hat{s}_{t,i} - s_{t,i}) \right|$. However, this objective has a complex, and generally nonconvex and nondifferentiable, dependence on the predictions $\hat{s}_t$, so directly optimizing it appears out of reach. Yet, we will now show how to achieve this via a two-layer algorithmic technique: first, a reduction to a surrogate minimax objective, followed by a "simulated play" solution of that minimax problem. For both layers, we will use OLO algorithms as subroutines.

**First Step:** We identify weights $w_t \in \Delta_{2dn}, t \in [T]$, in an online fashion such that the following surrogate objective $u = \sum_{t=1}^{T} u_t$ closely approximates $\Psi$:

$$\sum_{t=1}^{T} \overbrace{\sum_{\substack{i \in [d], j \in [n], \\ \sigma = \pm 1}} w_{t,(\sigma,i,j)} \cdot \sigma \cdot \vec{E}_j(x_t, \hat{s}_t) \cdot (\hat{s}_{t,i} - s_{t,i})}^{:= u_t(\hat{s}_t, s_t)}.$$

**Second Step:** While the surrogate function $\sum_t u_t$ usefully separates the original objective across rounds $t \in [T]$, each $u_t(\hat{s}_t, s_t)$ still depends on $\hat{s}_t$ through the event mappings, which need not be convex or differentiable. However, it is linear in the adversary's choice of $s_t$, and this can be exploited due to the following observation: If the adversary committed to $s_t$ first, the learner could achieve value 0 in the zero-sum game $\max_{s_t} \min_{\hat{s}_t} u_t(\hat{s}_t, s_t)$ by simply *copy-*

*ing* the adversary, i.e., with $\hat{s}_t = s_t$. Therefore, *simulating* the *reverse playthrough* of this game, with the adversary going first and the learner copying, can give us a randomized saddle-point strategy $\bar{s}_t$ for the learner: namely, the empirical distribution of the learner's simulated plays. This will suffice so long as the adversary plays to optimize the variable $s_t$ using *any* no-regret OLO algorithm. Therefore, by simulating sufficiently many rounds of this "no-regret adversary vs. copycat learner" game, we can get as close as we want to the value of the game.

Now we are ready to present our general Algorithm 1. For the first step, it instantiates the Chen et al. (2021) MsMwC algorithm for $2dn$ experts corresponding to signs $\sigma = \pm 1$, coordinates $i \in [d]$ and events $j \in [n]$, to enable bias bounds that depend on each event's incidence count. For the second step, it uses any no-regret OLO algorithm $\mathcal{A}_{\mathcal{S}}$ that can optimize over the state space $\mathcal{S}$; this could be OGD or any other general-purpose $O(\sqrt{T})$-regret algorithm.

---

**Algorithm 1** Unbiased Prediction

---

Initialize $\hat{s}_0 = \mathbf{0}^d$, $s_0 = \mathbf{0}^d$, $\vec{E}^0(\cdot) \equiv \mathbf{0}^n$, and $\mathcal{A}_{\text{MsMwC}}$.
**for** $t = 1 \ldots T$ **do**
  Get context $x_t \in \mathcal{X}$, and define $\vec{E}^t(\cdot) := \vec{E}(x_t, \cdot)$.
  Get new weights $w_t$ by updating $\mathcal{A}_{\text{MsMwC}}$ with losses:
  $\ell_{t-1}^{\text{outer}} \leftarrow \left( \sigma \cdot \vec{E}_j^{t-1} (\hat{s}_{t-1}) \cdot (s_{t-1,i} - \hat{s}_{t-1,i}) \right)_{\sigma,i,j}$
  Initialize a new instance of $\mathcal{A}_{\mathcal{S}}$ and any $s_0^{\text{tent}} \in \mathcal{S}$.
  **for** $\tau = 1 \ldots t^2$ **do**
    Get simulated prediction $s_\tau^{\text{tent}}$ by updating $\mathcal{A}_{\mathcal{S}}$ with:

$$\ell_{\tau-1}^{\text{inner}} \leftarrow \left( \sum_{\sigma=\pm 1} \sigma \sum_{j \in [n]} w_{t,(\sigma,i,j)} \cdot \vec{E}_j^t \left(s_{\tau-1}^{\text{tent}}\right) \right)_{i \in [d]}$$

  **end for**
  Set $\bar{s}_t \leftarrow \text{Unif}\left(\{s_0^{\text{tent}}, s_1^{\text{tent}}, \ldots, s_{t^2}^{\text{tent}}\}\right)$.
  Predict $\hat{s}_t \sim \bar{s}_t$.
  Observe true state $s_t$.
**end for**

---

**Theorem 2.4** (Bias of Algorithm 1). *For any time horizon $T$, and instantiated with any $O(\sqrt{T})$-regret OLO method $\mathcal{A}_{\mathcal{S}}$ over domain $\mathcal{S}$, Algorithm 1 produces (randomized) predictions $(\bar{s}_t)_{t\in[T]}$ whose realizations $(\hat{s}_t)_{t\in[T]}$ achieve the desired bias bounds for all $i \in [d]$ and $E_j, j \in [n]$:*

$$\text{Bias}_T(E, i) \leq O\left(\log(d|\mathcal{E}|T) + \sqrt{n_T(E)\log(d|\mathcal{E}|T)}\right).$$

*Proof.* **Step 1:** We instantiate $\mathcal{A}_{\text{MsMwC}}$ for $2dn$ experts, corresponding to signs $\sigma = \pm 1$, coordinates $i \in [d]$ and events $j \in [n]$. Let the weights of MsMwC be $(w_t)_{t \geq 1}$, and denote our loss vectors for MsMwC by $(\ell_t^{\text{outer}})_{t \geq 1}$, as defined in Algorithm 1. Denote experts' basis vectors by $e_{\sigma,i,j}$. For every $\sigma^* = \pm 1, i^* \in [d], j^* \in [n]$:

$$\text{Reg}_T\left(\mathcal{A}_{\text{MsMwC}}, e_{\sigma^*,i^*,j^*}\right) = \sum_{t\in[T]} \langle \ell_t^{\text{outer}}, w_t - e_{\sigma^*,i^*,j^*}\rangle$$

$$= \sum_{T,\sigma,i,j} w_{t,(\sigma,i,j)} \cdot \sigma \cdot \vec{E}_j^t\left(\hat{s}_t\right) \cdot (s_{t,i} - \hat{s}_{t,i})$$

$$+ \sigma^* \sum_{t=1}^{T} \vec{E}_{j^*}^t(\hat{s}_t) \cdot (\hat{s}_{t,i^*} - s_{t,i^*})$$

Rearranging and taking a max over $\sigma^*$, we get for all $i^*, j^*$:

$$\left| \sum_{t=1}^{T} \vec{E}_{j^*}^t(\hat{s}_t) \cdot (\hat{s}_{t,i^*} - s_{t,i^*}) \right|$$

$$\leq \max_{\sigma^* \in \pm 1} \text{Reg}_T(\mathcal{A}_{\text{MsMwC}}, e_{\sigma^*,i^*,j^*})$$

$$+ \sum_{T,\sigma,i,j} w_{t,(\sigma,i,j)} \cdot \sigma \cdot \vec{E}_j^t(\hat{s}_t) \cdot (\hat{s}_{t,i} - s_{t,i})$$

$$= O\left(\log(dnT) + \sqrt{\log(dnT) n_T\left(\vec{E}_{j^*}\right)}\right) + \sum_{t=1}^{T} u_t(\hat{s}_t, s_t),$$

where $u_t$ is as defined above, and the regret bound follows from Theorem 2.3 since the total squared loss of each expert $(\sigma, i, j)$ is: $\sum_{t=1}^{T} \left(\ell_{t,(\sigma,i,j)}^{\text{outer}}\right)^2 = O\left(\sum_{t=1}^{T} (\vec{E}_j^t(x_t, \hat{s}_t))^2\right) = O(n_T(\vec{E}_j))$.

**Step 2:** By definition of regret for $\mathcal{A}_{\mathcal{S}}$, we have for any $t$:

$$\text{Reg}_{t^2}(\mathcal{A}_{\mathcal{S}}) = \max_{s \in \mathcal{S}} \sum_{\tau \in [t^2]} \langle \ell_\tau^{\text{inner}}, s_\tau^{\text{tent}} - s \rangle$$

$$= \max_{s \in \mathcal{S}} \sum_{\tau \in [t^2]} \sum_{\sigma,i,j} w_{t,(\sigma,i,j)} \cdot \sigma \cdot \vec{E}_j^t\left(s_\tau^{\text{tent}}\right) \cdot \left(s_{\tau,i}^{\text{tent}} - s_i\right)$$

$$= \max_{s \in \mathcal{S}} \sum_{\tau \in [t^2]} u_t\left(s_\tau^{\text{tent}}, s\right) = t^2 \cdot \max_{s \in \mathcal{S}} \mathbb{E}_{\hat{s}_t \sim \bar{s}_t} [u_t(\hat{s}_t, s)].$$

The last line uses $\bar{s}_t \leftarrow \text{Unif}\left(\{s_0^{\text{tent}}, s_1^{\text{tent}}, \ldots, s_{t^2}^{\text{tent}}\}\right)$.

Thus, using that $\mathcal{A}_{\mathcal{S}}$ has regret $O(\sqrt{T})$, we have $\mathbb{E}_{\hat{s}_t \sim \bar{s}_t \forall t}\left[\sum_{t=1}^{T} u_t(\hat{s}_t, s_t)\right] \leq \sum_{t=1}^{T} t^{-2}\text{Reg}_{t^2}(\mathcal{A}_{\mathcal{S}}) = O(\sum_{t=1}^{T} t^{-1}) = O(\log T)$: a lower-order term. Taking the expectation of the Step 1 bound thus gives the result. $\square$

## 2.2. Efficient $O(T^{2/3})$ Online Multicalibration: Sketch

We now sketch a simple application illustrating that our high-dimensional prediction methodology can be useful even for single-dimensional forecasting. This application also showcases the utility of our per-event bias bounds scaling optimally as $O(\sqrt{n_T(E)})$ rather than as $O(\sqrt{T})$.

**Online Multicalibration** In this setting (Gupta et al., 2022; Hébert-Johnson et al., 2018), a learner, in each round

$t \in [T]$, receives a context $x_t \in \mathcal{X}$, makes (randomized) prediction $p_t \in [0,1]$, and receives true adversarial label $y_t \in [0,1]$. Upfront, a *group collection* $\mathcal{G} \subseteq 2^{\mathcal{X}}$ is specified, and the learner's goal is to minimize the expected calibration error (ECE) conditional on every group $G \in \mathcal{G}$: i.e. to minimize, for all groups $G \in \mathcal{G}$, the expectation of:[2]

$$ECE(G) = \sum_{p \in [0,1]} \left| \sum_{t=1}^{T} \mathbb{1}[x_t \in G] \mathbb{1}[p_t = p](y_t - p) \right|.$$

**Obtaining the $O(T^{2/3})$ Bound via Unbiased Prediction** We instantiate our framework by letting $P = (P_i)_{i \in [m]}$ be the $m$-point uniform discretization of $[0,1]$, and defining the following $m \cdot |\mathcal{G}|$ group-calibration events:

$$E_{G,i} = \mathbb{1}\left[ x_t \in G, p_t \in (P_i - \tfrac{1}{2m}, P_i + \tfrac{1}{2m}] \right].$$

For each "bucket" $(P_i \pm \frac{1}{2m})$, imagine reassigning all predictions $p_t$ that fell in this bucket to be $P_i$. For each group $G$, let $(n_{i,G})_{i,G}$ be the incidences of all discretized predicted values on $G$. We can then derive the bound:

$$ECE(G) = O(T/m) + \sum_{i \in [m]} O(\sqrt{n_{i,G}}),$$

where the first term is the discretization error, and the rest are bias bounds. In the worst case, this is $O(T/m + m\sqrt{T/m}) = O(T/m + \sqrt{Tm})$. By tuning $m = T^{1/3}$, we thus obtain *the first efficient $O(T^{2/3})$ online multicalibration method*. This rate in particular matches the rate of the algorithm of Abernethy et al. (2011) for vanilla calibration.

## 3. Unbiased Prediction for Decision Making

We now apply the unbiased framework to making predictions in the service of online adversarial decision making.

**Agents (Decision Makers)** We study agents (decision makers) who can choose amongst a set of actions $\mathcal{A} = \{1, \ldots, K\}$. They want to maximize utility as a function of both the action they take and of the state $s \in \mathcal{S} \subseteq \mathbb{R}^d$.

**Definition 3.1** (Agent's Utility). A utility function $u : \mathcal{A} \times \mathcal{S} \to [0,1]$ maps an action $a \in \mathcal{A}$ and a state $s \in \mathcal{S}$ to $u(a, s)$. We assume that for every action $a \in \mathcal{A}$, $u$ is *linear* and *L-Lipschitz* in $s$, so that $|u(a, s_1) - u(a, s_2)| \leq L \|s_1 - s_2\|_\infty$ for all $s_1, s_2 \in \mathcal{S}$ and some $L > 0$.

**Definition 3.2** (Best-Response). The *best response function*[3] $\mathrm{BR}_u : \mathcal{S} \to \mathcal{A}$ for utility $u$ is: $\mathrm{BR}_u(s) = \operatorname*{argmax}_{a \in \mathcal{A}} u(a, s)$.

Suppose we make predictions $\hat{s}_1, \ldots, \hat{s}_t$. An agent with utility $u$ may use them to take corresponding actions $a_1, \ldots, a_t$.

---

[2]The summation over $p \in [0,1]$, despite looking uncountable, only has $T$ nonzero terms, corresponding to $p \in \{p_1, \ldots, p_T\}$.

[3]We assume that all ties are broken lexicographically.

We call an agent *straightforward* if they trust the predictions as correct (as if $s_t = \hat{s}_t$) and thus always best respond:

**Definition 3.3** (Straightforward Agent). An agent with utility $u$ who treats predictions as correct and on every round $t$ chooses $a_t = \mathrm{BR}_u(\hat{s}_t)$ is called *straightforward*.

**Regret** Since our predictions need not be correct, a straightforward agent may regret not having taken some other sequence of actions in hindsight (i.e. with knowledge of the true states $s_1, \ldots, s_t$). We study several regret notions.

**Definition 3.4** (External regret). The external regret of a utility-$u$ agent is defined as:

$$\mathrm{Reg}_T(u) := \max_{a \in \mathcal{A}} \sum_{t=1}^{T} u(a, s_t) - u(a_t, s_t).$$

**Definition 3.5** (Swap regret). A mapping $\phi : \mathcal{A} \to \mathcal{A}$ is called a strategy modification mapping. Let $\Phi$ be the set of all such mappings. The *swap regret* of a utility-$u$ agent is:

$$\mathrm{SwapReg}_T(u) := \max_{\phi \in \Phi} \sum_{t=1}^{T} u(\phi(a_t), s_t) - u(a_t, s_t).$$

External regret compares the agent's play to the best fixed action. Swap regret (Blum & Mansour, 2007) is strictly more challenging (indeed, external regret is equivalent to competing against the $K$ constant strategy modification functions $(\phi_a)_{a \in \mathcal{A}}$, where $\phi_a : \mathcal{A} \to \mathcal{A}$ is given by $\phi_a(a') = a$ for $a' \in \mathcal{A}$), and allows the agent to compete against all re-mappings of their actions into other actions.

We now introduce the strong notion of *conditional regret*, parameterized by collections of events that may depend on the contexts and on the agent's actions. It requires the agent to have no external regret conditional on every event.

**Definition 3.6** (Conditional Regret (Lehrer, 2003; Blum & Mansour, 2007; Lee et al., 2022)). Fix $\Xi$, a finite collection of covariate-dependent and action-dependent subsequences of rounds: each member $\xi \in \Xi$ is a mapping $\mathcal{X} \times \mathcal{A} \to [0,1]$. The $\Xi$-*conditional regret* of a utility-$u$ agent is:

$$\mathrm{CReg}_T(\Xi, u)$$
$$= \max_{\xi \in \Xi, a \in \mathcal{A}} \sum_{t=1}^{T} \xi(x_t, a_t) \left( u(a, s_t) - u(a_t, s_t) \right).$$

### 3.1. Swap Regret Guarantees for Many Agents

**Environment** Consider any convex compact state space $\mathcal{S} \subseteq \mathbb{R}^d$. Suppose there are $n$ agents, each with $K$ discrete actions and with utility functions $(u_i)_{i \in [n]}$ that are linear and $L$-Lipschitz in the state variable $s \in \mathcal{S}$. We will now show how to make predictions $(\hat{s}_t)_{t \in [T]}$ to *simultaneously* guarantee no swap regret to every agent, given

that all agents are straightforward, i.e. they all best-respond: $a_{t,i} = \mathrm{BR}_{u_i}(\hat{s}_t)$ for $i \in [n]$. We will do it by applying the unbiased prediction framework with the following natural collection of $nK$ events.

**Best-Response Events** We write $E_{u,a}(s) = \mathbb{1}[\mathrm{BR}_u(s) = a]$ to denote the binary event that $a$ is a best response to $s$ for utility $u$. These events are essentially *level-set* events of the agent's best-response correspondence. We will now see that producing $\mathcal{E}$-conditionally unbiased predictions $(\hat{s}_t)_{t \in [T]}$ for the $nK$-sized event collection $\mathcal{E} = (E_{u_i,a})_{i \in [n], a \in [K]}$ will suffice to guarantee no swap regret to all agents.

Informally, the reason these events give no swap regret to each agent $u_i$ is the following. Fix any strategy modification function $\phi \in \Phi$. Then each event $E_{u_i,a}$ will ensure that on those rounds $t$ where $u_i$ played $a \in [K]$, the predictions $\hat{s}_t$ are sufficiently unbiased that $a = \mathrm{BR}_{u_i}(\hat{s}_t)$, the best response that assumes the predictions are correct, is in fact the (approximately) best action to play over those rounds; in particular, $a$ will have no regret to the re-mapped action $\phi(a) \in [K]$ on those rounds. Since this argument applies to all re-mappings $\phi \in \Phi$ and all actions $a \in [K]$, it will by definition ensure no swap regret to agent $u_i$.

Formally, fix any $u_i$ and swap $\phi : \mathcal{A} \to \mathcal{A}$. We express the agent's regret to swap $\phi$ in terms of $(E_{u_i,a})_{a \in [K]}$ as:

$$\sum_{t=1}^{T} u_i(\phi(a_{t,i}), s_t) - u_i(a_{t,i}, s_t)$$

$$= \sum_{a \in \mathcal{A}} \sum_{t:\mathrm{BR}_{u_i}(\hat{s}_t)=a} u_i(\phi(a), s_t) - u_i(a, s_t)$$

$$= \sum_{a \in \mathcal{A}} \sum_{t=1}^{T} E_{u_i,a}(\hat{s}_t) \left( u_i(\phi(a), s_t) - u_i(a, s_t) \right).$$

By linearity of $u_i$ in $s_t$, we combine terms to get:

$$\sum_{a \in \mathcal{A}} u_i \left( \phi(a), \sum_{t=1}^{T} E_{u_i,a}(\hat{s}_t) s_t \right)$$

$$- \sum_{a \in \mathcal{A}} u_i \left( a, \sum_{t=1}^{T} E_{u_i,a}(\hat{s}_t) s_t \right).$$

Now, by $L$-Lipschitzness of $u_i$, for $a' \in \{a, \phi(a)\}$ we have

$$\mathbb{E} \left| u_i \left( a', \sum_{t=1}^{T} E_{u_i,a}(\hat{s}_t) \hat{s}_t \right) - u_i \left( a', \sum_{t=1}^{T} E_{u_i,a}(\hat{s}_t) s_t \right) \right|$$

$$\leq L \cdot \mathbb{E} \left\| \sum_{t=1}^{T} E_{u_i,a}(\hat{s}_t) \cdot (\hat{s}_t - s_t) \right\|_{\infty} = L \cdot \mathrm{Bias}_T(E_{u_i,a}).$$

Applying this to both terms in the above $\phi$-regret expression, we have that it is only an ex-

pected $L \sum_{a \in [K]} \mathrm{Bias}_T(E_{u_i,a})$ error away from:

$$\sum_{a \in \mathcal{A}} \left( u_i \left( \phi(a), \sum_{t=1}^{T} E_{u_i,a}(\hat{s}_t) \hat{s}_t \right) - u_i \left( a, \sum_{t=1}^{T} E_{u_i,a}(\hat{s}_t) \hat{s}_t \right) \right).$$

However, this last expression can be rewritten as $\sum_{t=1}^{T} u_i(\phi(a_{t,i}), \hat{s}_t) - u_i(a_{t,i}, \hat{s}_t)$, which is nonpositive since actions $a_{t,i} = \mathrm{BR}_{u_i}(\hat{s}_t)$ obtain the best utility *when evaluated on predicted states* $\hat{s}_t$. This means that we have shown a $L \sum_{a \in [K]} \mathrm{Bias}_T(E_{u_i,a})$ expected regret bound for any agent to any swap function $\phi : \mathcal{A} \to \mathcal{A}$, which implies that each agent has expected swap regret at most $L \sum_{a \in [K]} \mathrm{Bias}_T(E_{u_i,a})$.

Note that exactly one event from $\{E_{u_i,a}\}_{a \in \mathcal{A}}$ occurs at each time $t$. Thus, $\sum_{a \in \mathcal{A}} n_T(E_{u,a}) \leq T$, so that $L \sum_{a \in [K]} \mathrm{Bias}_T(E_{u_i,a}) = O(L \sum_{a \in [K]} \sqrt{n_T(E_{u_i,a})}) \leq O(LK\sqrt{T/K}) = O(L\sqrt{KT})$. Therefore, we have shown:

**Theorem 3.7** (No Swap Regret for Multiple Agents)**.** *In the above setting with $n$ agents, all with $K$ actions and Lipschitz utilities, if all agents best-respond to our forecasts $(\hat{s}_t)_{t \in [T]}$, then by making these forecasts via $\mathcal{E}$-unbiased prediction for $\mathcal{E} = (E_{u_i,a})_{i \in [n], a \in [K]}$, the we efficiently obtain $O(\sqrt{KT})$ swap regret bounds for all agents simultaneously.*

In the context of experts learning for a single decision maker, similar observations about the relevance of best-response partitions have been previously made by Perchet (2011) and Haghtalab et al. (2023b).

## 4. Conditional Regret Guarantees for Online Combinatorial Optimization

The regret guarantees that we just obtained for $n$ agents with size-$K$ action sets required unbiased predictions conditional on $O(nK)$ events, resulting in $poly(nK)$ runtime. This method applies to any finite action sets, and will be efficient where agents' action sets are modestly sized. However, when the agents' action sets are combinatorially large, this runtime is prohibitive. Below we identify an important setting in which an exponentially improved (oracle-) complexity $poly(n \log K)$ can be obtained.

### 4.1. Setting: Online Combinatorial Optimization

In a combinatorial optimization setting (as studied by Kalai & Vempala (2005)), there are $d \geq 1$ *base elements*, or *base actions*, $e \in B := \{1, \ldots, d\}$, each offering an associated reward $r_e \in [-1, 1]$. In this setting, an agent has action space $\mathcal{A} \subseteq 2^B$ — an arbitrarily structured collection of subsets of the base action set — and their utility $u : \mathcal{A} \times \mathcal{S} \to [-d, d]$ is defined as the sum of the rewards of the base actions in the chosen action:

$$u(a, r) := \sum_{e \in a} r_e \quad \text{for } a \in \mathcal{A}, r \in \mathcal{S} := [-1, 1]^d.$$

Thus, given a vector $r = (r_e)_{e \in [d]}$ of $d$ base rewards, the agent's optimization task is to identify, from among their actions $a \in \mathcal{A}$, the highest-reward subset of the base action set. An *offline oracle* for this problem is any algorithm that, given $r$, computes the agent's best action in $\mathcal{A}$.

Now, we define a contextual online $n$-agent setting in which base rewards vectors $r_t$ are generated by an adversary in rounds $t \in [T]$. In this setting, each agent $i$'s goal will be to learn to play actions $(a_{t,i})_{t \in [T]}$ that will minimize an appropriate notion of regret to the hindsight-best policy from some benchmark policy class.

Formally, consider an arbitrary context space $\mathcal{X}$, and $n \geq 1$ combinatorial agents with action sets $(\mathcal{A}_i)_{i \in [n]}$ repeatedly playing the following game in rounds $t \in [T]$. In round $t$:

1. Agents $i \in [n]$ observe context $x_t \in \mathcal{X}$, and commit to their actions $a_{t,i} \in \mathcal{A}_i$;

2. Adversary produces base rewards vector $r_t \in [-1, 1]^d$;

3. Agents see $r_t$ and get utilities $u_i(a_{t,i}, r_t) := \sum_{e \in a_{t,i}} r_{t,e}$.

**Examples** Suppose the base elements $B$ correspond to the roads in a road network, the feasible subsets $\mathcal{A}_i$ for each agent $i \in [n]$ correspond to collections of roads that form source-to-sink paths for that agent in the network, and the reward for each road (edge) $e \in B$ in the network is the (negative) latency on this edge. This classic instance of online combinatorial optimization is called online routing or online shortest paths (Takimoto & Warmuth, 2003; Kalai & Vempala, 2005). More generally, the action spaces $\mathcal{A}_i$ could represent *any* combinatorial structure; other well-studied examples include spanning trees, Hamiltonian paths, and fixed-size subsets of the base set. For many of these classical examples, there exist efficient offline oracles, such as Bellman-Ford for shortest paths or Prim or Kruskal for spanning trees.

**Regret Guarantees** The FTPL algorithm of Kalai & Vempala (2005) reduces the problem of obtaining efficient *external* regret bounds for combinatorial optimization problems to the offline problem of linear optimization over the action spaces $\mathcal{A}_i$. Here we show for the first time how to efficiently obtain much stronger and more granular regret bounds: namely, $\Xi$-*conditional* regret bounds for any polynomially large collection of events $\Xi$. Moreover, unlike prior results, our result will provide these guarantees simultaneously for any finite collection of agents, letting us publish a concise forecast that is simultaneously useful for many downstream consumers.

Some existing general-purpose online algorithms (Blum & Mansour, 2007; Lee et al., 2022; Haghtalab et al., 2023a) could be used to obtain conditional regret bounds sublinear

in $T$, by directly optimizing over the entire action set $\mathcal{A}_i$ of each agent. However, their runtime will then scale polynomially in $|\mathcal{A}_i|$ which can be as large as $\Omega(2^d)$, thus making the runtime exponentially large in the problem size.

Our framework, by contrast, will let us give the agents conditional regret guarantees simply by unbiasedly predicting the $d$-dimensional base reward vectors (conditionally on appropriate events). Our general algorithm will thus run in time $poly(d)$, giving us an efficient algorithm with an exponential runtime improvement compared to prior work.

### 4.2. Conditional Regret via Unbiased Prediction

To derive conditional regret guarantees via unbiased prediction, we will use the same Predict-then-Act approach as in Section 3. At the beginning of each round $t$, we will (appropriately unbiasedly) predict the rewards vector $\hat{r}_t \in \mathcal{S}$. Every agent $i \in [n]$ will then best-respond to our prediction and select action $a_{t,i} = \mathrm{BR}_{u_i}(\hat{r}_t)$.

However, to make this approach efficient, we must now design our event collection differently than before. Indeed, consider the simplest case where we ask for no external regret to downstream agents. The collection of "level set" events $\{E_{u_i,a_i}\}_{i \in [n], a_i \in \mathcal{A}_i}$ studied in Section 3 will imply sublinear regret as before — but will be too big as it scales with $|\mathcal{A}_i|$, which can be exponential in $d$.

To overcome this, we will take advantage of the special structure of the payoffs, which are all linear in the base element rewards. The idea is to condition on events defined by the *base elements* $e \in B = [d]$. Again starting with no external regret, it turns out that for each agent $i \in [n]$ it suffices to condition on the always-on event, and on $d$ events $(E_e)_{e \in [d]}$: for each base element $e$, $E_e$ will be the event that *the agent's chosen action $a_{t,i}$ contains $e$*, i.e., $E_e(s) = 1[e \in \mathrm{BR}_{u_i}(s)]$. This requires just $nd$ events over all agents. From here, if we now desire $\Xi$-conditional regret guarantees, it will suffice to expand this event collection to $O(|\Xi| \cdot d)$ events per agent: for each $\xi \in \Xi$, the intersectional events $E_{e,\xi}(x, s) = \xi(x, s) \cdot E_e(s)$ for $e \in [d]$ (that both $\xi$ is active and $E_e$ is active), as well as the event that is active whenever $\xi$ is active, will imply no external regret conditional on $\xi$.

Importantly, observe that all these events can be evaluated via direct calls to the offline optimization oracle for the problem (e.g., Bellman-Ford for routing); therefore, the unbiased prediction algorithm will be oracle efficient. We now state and prove our conditional regret bound.

**Theorem 4.1.** *Consider online combinatorial optimization over context space $\mathcal{X}$ with $d$ base actions and $n$ agents with action sets $\mathcal{A}_i \subseteq 2^{[d]}$. Suppose each agent $i$ is straightforward, and wants to obtain no $\Xi_i$-conditional regret for some*

events $\Xi_i$. Define the following set of events:

$$\mathcal{E} = \bigcup_{i \in [n]} \bigcup_{\xi \in \Xi_i} \left\{ \left\{ E_{e,\xi}^i \right\}_{e \in [d]} \cup \left\{ E_\xi^i \right\} \right\}.$$

Here, $E_{e,\xi}^i(x_t, \hat{r}_t) := \mathbb{1}[e \in \mathrm{BR}_{u_i}(\hat{r}_t)] \cdot \xi(x_t, \mathrm{BR}_{u_i}(\hat{r}_t))$; and $E_\xi^i(x_t, \hat{r}_t) := \xi(x_t, \mathrm{BR}_{u_i}(\hat{r}_t))$, for all $e, i, \xi \in \Xi_i$.

Then, running Unbiased Prediction on state space $\mathcal{S} = [-1,1]^d$ with event collection $\mathcal{E}$ will produce a sequence of predictions $(\hat{r}_t)_{t \in [T]}$ such that each agent $i \in [n]$, by playing their best-response actions $a_{t,i} = \mathrm{BR}_{u_i}(\hat{r}_t)$ at all rounds, will obtain $\widetilde{O}(d\sqrt{T})$ expected $\Xi_i$-conditional regret. The runtime will consist of $poly\left(dT \sum_{i=1}^{n} |\Xi_i|\right)$ oracle calls to the offline optimization oracle for the setting.

*Proof.* It suffices to fix any agent $i \in [n]$ and event $\xi \in \Xi_i$ and show that best-responding to the $\mathcal{E}$-unbiased predictions $(\hat{r}_t)_{t \in [T]}$ gets $i$ no external regret on subsequence $\xi$. Denote the agent's external regret on the subsequence $\xi$ by:

$$\mathrm{CReg}_T(\xi, i)$$
$$= \max_{a^* \in \mathcal{A}_i} \sum_{t=1}^{T} \xi(x_t, a_{t,i}) \cdot (u_i(a^*, r_t) - u_i(a_{t,i}, r_t)).$$

Now, consider the hypothetical "ideal" scenario in which our predictions are exactly correct on every round, i.e., $\hat{r}_t = r_t$ for all $t$. Then, our "ideal" regret would be nonpositive:

$$\mathrm{IdReg}_T(\xi, i)$$
$$= \max_{a^* \in \mathcal{A}_i} \sum_{t=1}^{T} \xi(x_t, a_{t,i}) \cdot (u_i(a^*, \hat{r}_t) - u_i(a_{t,i}, \hat{r}_t)) \le 0$$

since for each $t$, $a_{t,i} = \mathrm{BR}_{u_i}(\hat{r}_t)$ and thus $u_i(a_{t,i}, \hat{r}_t) = \max_{a \in \mathcal{A}_i} u_i(a, \hat{r}_t) \ge u_i(a^*, \hat{r}_t)$. Therefore,

$$\mathrm{CReg}_T(\xi, i) \le \mathrm{CReg}_T(\xi, i) - \mathrm{IdReg}_T(\xi, i).$$

This difference in regrets can be expressed as:

$$\max_{a^* \in \mathcal{A}_i} \sum_{t=1}^{T} \xi(x_t, a_{t,i}) \cdot u_i(a^*, r_t)$$
$$- \max_{a^* \in \mathcal{A}_i} \sum_{t=1}^{T} \xi(x_t, a_{t,i}) \cdot u_i(a^*, \hat{r}_t)$$
$$+ \sum_{t=1}^{T} \xi(x_t, a_{t,i}) \cdot (u_i(a_{t,i}, \hat{r}_t) - u_i(a_{t,i}, r_t)).$$

It is easy to check that the first line's expectation is at most $\mathrm{Bias}_T(E_\xi^i)$. Similarly, by decomposing the second line's expression across the $d$ coordinates, it can be seen that its expectation is at most $\sum_{e \in [d]} \mathrm{Bias}_T(E_{e,\xi}^i)$. Hence, the expected regret of agent $i$ conditional on $\xi$ is at most $\sum_{e \in [d]} \mathrm{Bias}_T(E_{e,\xi}^i) + \mathrm{Bias}_T(E_\xi^i) = \widetilde{O}(d\sqrt{T})$. $\quad\square$

## Acknowledgements

We are grateful to Edgar Dobriban, Amy Greenwald, Jason Hartline, Michael Jordan, Shuo Li, Jon Schneider, and Rakesh Vohra for insightful conversations at various stages of this work.

## Impact Statement

This paper presents work whose goal is to advance the field of Machine Learning. There are many potential societal consequences of our work, none which we feel must be specifically highlighted here.

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

# A. Additional Related Work

**Calibration**   The study of sequential calibration goes back to (Dawid, 1985) who viewed it as a way to define the foundations of probability, and algorithms for producing calibrated forecasts in an adversarial setting were first given by (Foster & Vohra, 1998). (Foster & Vohra, 1999) were the first to connect sequential calibration to sequential decision making, showing that a decision maker who best responds to (fully) calibrated forecasts obtains diminishing internal regret (and that when all agents in a game do so, empirical play converges to correlated equilibrium). (Kakade & Foster, 2008) and (Foster & Hart, 2018) make a similar connection between "smooth calibration" (which in contrast to classical calibration can be obtained with deterministic algorithms) and Nash equilibrium.

Much of the calibration literature has focused on the binary prediction and one dimensional regression settings, where labels are in $\{0, 1\}$ or in $[0, 1]$, and predictions are in $[0, 1]$. Comparatively few works, including (Zadrozny & Elkan, 2002; Kuleshov & Liang, 2015; Gupta & Ramdas, 2021), have addressed higher dimensional predictions, which, as discussed, are challenging because of the curse of dimensionality; many of these works have sought to *reduce* multiclass calibration problems to binary calibration. In this context, our work in particular proposes tractable notions of online multiclass calibration for cases when there is a specific downstream task that the forecasts will be used for.

**Multicalibration**   In the recent computer science literature, there has been interest in constructive calibration guarantees (obtained by efficient algorithms and obtaining good rates) that hold conditional on context in various ways, called *multicalibration* (Hébert-Johnson et al., 2018). Multicalibration has been studied both in the batch setting (Hébert-Johnson et al., 2018; Kim et al., 2019; Globus-Harris et al., 2023; Haghtalab et al., 2023a) and in the online sequential setting (Foster & Kakade, 2006; Foster et al., 2011; Gupta et al., 2022; Garg et al., 2024). For the most part (with a few notable exceptions (Gopalan et al., 2022b; Zhao et al., 2021)) multicalibration has been studied in the 1-dimensional setting in which the outcome being predicted is boolean. This has been extended to predicting real-valued outcomes, with notions of calibration tailored to variances (Jung et al., 2021), quantiles (Bastani et al., 2022; Jung et al., 2023), and other distributional properties (Noarov & Roth, 2023). See (Roth, 2022) for an introductory exposition of this literature. Our algorithm can be used to recover many of the above online multicalibration guarantees by plugging in appropriate events, but it goes beyond multicalibration constraints.

**Omniprediction**   A growing line of work (Gopalan et al., 2022a; 2023a;b; 2024) aims to use (multi)calibration as a tool for a one-dimensional form of downstream decision making, called omniprediction. The goal of omniprediction is to make probabilistic predictions of a binary outcome as a function of contextual information that are useful for simultaneously optimizing a variety of downstream loss functions. E.g., (Gopalan et al., 2022a) show that a predictor that is multicalibrated with respect to a benchmark class of functions $\mathcal{H}$ and of a binary label can be used to optimize any convex, Lipschitz loss function of an action and a binary label. Also related is the *outcome indistinguishability* strand of research (Dwork et al., 2021; 2022), which studies producing decisions that are indistinguishable from the ground truth according to a collection of tests.

Conceptually, our motivation is slightly different than for omniprediction: while omniprediction aims to produce forecasts that are good enough to optimize for a *large* (typically infinitely large) family of *possible* downstream tasks characterized by their associated losses — such that we may not know ahead of time which task will present itself — our framework is developed to handle finitely many arbitrary but *specific* (i.e., known-in-advance) downstream tasks. The above mentioned results are in the batch setting.

In the online setting, (Kleinberg et al., 2023) defined "U-calibration", which can be viewed as a non-contextual version of omniprediction where the goal is to make predictions that guarantee an arbitrary downstream decision maker no external regret. In comparison to (Kleinberg et al., 2023), our goal is to give both stronger guarantees than external regret, and to be able to do so even when the state space is very large.

**Calibration for Decision Making**   The most closely related work is (Zhao et al., 2021), who define and study "decision calibration" in the batch setting in the context of predicting a probability distribution over $k$ discrete outcomes. Decision calibration is a slightly weaker requirement than what we study, also defined in terms of the best-response correspondence of a decision maker's utility function. Decision calibration asks, informally, that a decision maker be able to correctly estimate the expected reward of their best response policy; we ask for a slightly stronger condition that requires them to also be able to estimate the utility of deviations as a function of their play. This kind of unbiased estimation (based on the best-response correspondence of a decision maker) has also been previously observed to be related to swap regret in (Perchet, 2011) and

(Haghtalab et al., 2023b). The algorithmic portion of our work can be viewed as extending (Zhao et al., 2021) from the batch to the online adversarial setting; Our applications hinge crucially on both the online aspect of our algorithm and on the more general setting we consider, beyond predicting distributions on $k$ outcomes.

Subsequently to (Zhao et al., 2021), decision calibration has been extended to, or applied in, several specific downstream tasks in the batch setting. For instance, (Fisch et al., 2022; Wang et al., 2022) applied decision calibration in the presence of downstream selection or screening processes. These and omniprediction ideas were also used to obtain new *performative prediction* algorithms in (Kim & Perdomo, 2023). In the opposite direction, (Rothblum & Yona, 2022) study how downstream decision policies can be modified in response to miscalibrated forecasts.

**Predict then Optimize**    An expansive recent literature has focused on the similarly named *predict-then-optimize* problem (Elmachtoub & Grigas, 2022; El Balghiti et al., 2019; Liu & Grigas, 2021). This line of work investigates a setup in which predictions made from data are to be used in a linear optimization problem downstream in the pipeline. This is similar in motivation to our framework, but with two important differences: (1) the predict-then-optimize framework aims to optimize for a single downstream problem, whereas we aim to simultaneously provide guarantees to an arbitrary finite collection of downstream decision makers; and (2) the surrogate loss approach studied in this literature is naturally embedded in a batch/distributional setting, where the goal is to exactly optimize for the Bayes optimal downstream decision policy, up to generalization/risk bounds; meanwhile, our framework naturally lives in the online adversarial setting, and aims for different notions of optimality defined in terms of regret bounds, as well as omniprediction-type 'best-in-class' optimality. Both frameworks can be used to solve downstream combinatorial optimization problems (Mandi et al., 2020; Demirović et al., 2019); but our framework appears to have a broader set of applications — as a consequence of its strong calibration properties, we are able to apply our framework to derive strong uncertainty quantification guarantees, which do not appear to naturally fit within the predict-then-optimize framework. There also exist other approaches for learning in batch decision making pipelines, that are different from the predict-then-optimize method; see e.g. (Donti et al., 2017; Khalil et al., 2017; Wilder et al., 2019; Vanderschueren et al., 2022).

## B. Calibration and Decision Making

The notion of calibration (Dawid, 1985) requires that predictors make forecasts that are consistent with the ground truth conditional on the predicted values themselves. For instance, for a binary predictor $f : \mathcal{X} \to [0, 1]$, it enforces, roughly speaking, that $\mathbb{E}_{(x,y)}[y|f(x) \approx v] \approx v$, for all $v \in [0, 1]$.

Calibration has very strong decision-theoretic properties. When making predictions about a payoff-relevant state, in a very general setting it is a dominant strategy amongst all *prediction-to-action policies* for every downstream decision maker to best-respond to calibrated predictions as if they were correct. This has strong semantics as "trustworthiness" — as one can do no better than to trust calibrated predictions and *act accordingly*. It also implies strong performance guarantees for the downstream decision makers. Here are two examples: First, decision makers who best respond to calibrated forecasts are guaranteed to have no swap regret — meaning that they obtain utility that is as high as the best action they could have played in hindsight, not just marginally, but also *conditionally* on each action that they played (Foster & Vohra, 1999). The second example concerns multi-class prediction, where a decision-maker observes features and predicts an unknown label from some large set. Standard machine learning methods for multi-class classification will, given features, predict "scores" for each label that look like probabilities in that they are non-negative and sum to 1. These scores are not probabilities; nevertheless, decision makers who produce "prediction sets" of labels by treating *calibrated* scores as if they were real probabilities will find that their prediction sets cover the true label with the same frequency that they would if the scores really were conditional label probabilities. So sequential calibration offers very strong guarantees — and it has been known since (Foster & Vohra, 1998) that it is possible to produce calibrated forecasts even in adversarial environments.

But calibration is in different senses both too strong and too weak. On the one hand, calibration is too weak in that it provides only a *marginal* guarantee; calibrated forecasts will in general fail to be calibrated conditional on external information. Thus, the property that downstream agents can do no better than to treat calibrated forecasts as correct will fail to hold if the downstream agents have access to external context. In one-dimensional prediction settings (such as real-valued regression), *multicalibration* (Hébert-Johnson et al., 2018) mitigates this weakness: it allows one to enforce calibration not just marginally, but also conditionally on any collection of context-dependent *groups* or *subpopulations* in the data.

On the other hand, calibration is too strong in that (because it is agnostic to downstream decisions) it conditions on fine distinctions in its predictions that may be irrelevant to the downstream task at hand. As a result, calibration is intractable in

*high-dimensional* settings. Since calibrated predictions must be statistically unbiased conditional on their own values, then for $d$-dimensional prediction problems — in which, up to discretization, there are $\Omega(2^d)$ possible values we may predict — we, naively, need to respect $\Omega(2^d)$ possible conditioning events to stay calibrated. Given this intuition, it should come as no surprise that the best known calibration algorithms have exponential computational and statistical complexity in dimension $d$ of the outcome space; there exist some lower bounds in the literature that confirm this hardness, see e.g. the PPAD-hardness result of (Hazan & Kakade, 2012)). In fact, even in 1 dimension, it is known that achieving adversarial calibration at a rate of $O(\sqrt{T})$ is impossible (Qiao & Valiant, 2021) — even though it is possible to obtain swap regret at this rate (Blum & Mansour, 2007). Thus, despite its remarkable guarantees, calibration has been of little utility in designing online algorithms for high-dimensional problems.

## C. Connecting Prediction and Decision Making

We next make connections between our ability to make unbiased predictions and the quality of decisions that are made downstream as a function of our predictions in a general setting. The form of the argument will proceed in the same way that it will in our main applications, and so is instructive.

Specifically, we will show that a straightforward decision maker who simply best responds to our predictions can be guaranteed *no swap regret* — if we just make our predictions *unbiased* conditional on the events defined by the decision maker's best-response correspondence.

**The Predict-then-Act Paradigm**   These results, whose formal statements are given below, suggest a natural design paradigm for sequential decision algorithms, which we call *predict-then-act*. The idea is simple: first we make a prediction $\hat{s}_t$ for an unknown payoff-relevant parameter $s_t$, and then we choose an action as if our prediction were correct — i.e. we best respond to $\hat{s}_t$. We can parameterize the predict-then-act algorithm with various events $\mathcal{E}$, such that our predictions will be unbiased with respect to events in $\mathcal{E}$. Whenever $\mathcal{E}$ is a collection of polynomially many events that can each be evaluated in polynomial time, the predict-then-act algorithm can be implemented in polynomial time per step. While Predict-Then-Act is quite simple, its flexibility in a variety of settings lies in the design of the event set $\mathcal{E}$ and prediction space $\mathcal{S}$. By choosing the events $\mathcal{E}$ to be appropriately tailored to the task at hand, we can arrange that Predict-Then-Act has guarantees of various sorts.

---

**Algorithm 2** `Predict-Then-Act`$(T, \mathcal{U}, \mathcal{E}, \mathcal{S}, \mathcal{A})$

  **for** $t$ in $1 \ldots T$ **do**
    Compute $\psi_t \leftarrow$ `UnbiasedPrediction`$(\mathcal{E}, t)$
    Predict $\hat{s}_t \sim \psi_t$
    **for** $u_i \in \mathcal{U}$ **do**
      Decision maker $i$ selects action $a_{t,i} = \mathrm{BR}_{u_i}(\hat{s}_t) = \operatorname{argmax}_{a \in \mathcal{A}} u_i(a, \hat{s}_t)$
    **end for**
    Observe outcome $s_t \in \mathcal{S}$
  **end for**

---

We now develop, and use, our machinery based on the Predict-then-Act approach powered by our Unbiased Prediction algorithm's guarantees, to give algorithms with strong no-regret guarantees in a variety of sequential settings. In all cases, the scenario we analyze is that in rounds $t$, a predictor makes state predictions $\hat{s}_t$, after which one or more decision makers choose actions that are best responses to $\hat{s}_t$ (i.e. they function as *straightforward* decision makers). When we are designing an algorithm for a single decision maker, we always use the *predict-then-act* paradigm (Algorithm 2). When we are designing a coordination mechanism, we imagine that our predictions are simultaneously issued, on every round, to multiple decision makers, who each independently act. We show how guaranteeing that our predictions are unbiased subject to appropriately chosen events gives desirable guarantees for downstream decision makers of various sorts.

### C.1. Transparent Policy Evaluation and Its Downstream Benefits

In this subsection, we introduce the main benefit of calibration that underlies, in one form or another, all our following applications. Namely, calibrated (or, as we will see, sufficiently unbiased) predictions result in what we call a *transparent policy evaluation* property, whereby downstream agents' prediction-to-action policies will in hindsight (once the true states

are revealed) bring as much utility to the agents as they would get *had the predictions been exactly correct*. In that sense, the predicted states are *indistinguishable* from true states *for the purposes of policy evaluation*. In a nutshell, this enables straightforward downstream optimization of the next action to play while only having access to predicted, rather than realized, quantities — and is the main vehicle that drives our Predict-then-Act approach where agents simply best-respond to the predictions. This general "transparency" property of calibration is also what underlies outcome indistinguishability (Dwork et al., 2021) and omniprediction (Gopalan et al., 2023a) in 1-dimensional batch settings, and (Zhao et al., 2021) gave similar transparency guarantees in multi-dimensional batch settings — but as we will see, will be especially useful for us in problems arising in high-dimensional online settings.

In this and the next subsection, we begin by deriving this transparency property (and its downstream consequences) in the idealized scenario where the predictions $\hat{s}$ are *exactly* calibrated (i.e., there is no error term). This, of course, is a statistically unachievable goal in high dimensions, and so these results won't yet be implementable. However, this presents no issue for our algorithmic results — which do have error rates — because given access to approximately calibrated or unbiased predictions (which we will generally obtain by invoking our Unbiased Prediction algorithm), bias error propagation through our idealized proof templates is quite easy. Meanwhile, these idealized statements and their proofs provide the core statistical intuition about why our approach works.

We begin by formally defining how we want to evaluate the long-term success of any prediction-to-action policy in our online setting.

**Definition C.1** (Policy Evaluation Function). A policy evaluation function is a mapping $U : (\mathcal{S} \to \mathcal{A}) \times \mathcal{S}^* \times \mathcal{S}^* \to \mathbb{R}$. The interpretation is that for any prediction-to-action policy $f : \mathcal{S} \to \mathcal{A}$ and any two sequences of states $s, \hat{s} \in \mathcal{S}^*$ with $\text{len}(s) = \text{len}(\hat{s}) < \infty$, $U(f, \hat{s}, s)$ will give the total utility of the policy $f$ evaluated on the ground truth state sequence $s$ when actions are taken according to the predicted state sequence $\hat{s}$.

We will usually instantiate this definition as follows. Fixing any time horizon $T$ and any decision maker with utility function $u : \mathcal{A} \times \mathcal{S} \to \mathbb{R}$, we will let the evaluation function $U$ be defined as the decision maker's cumulative (total) utility of employing the policy $f$ applied to predictions $\hat{s}$ across rounds $1, \ldots, T$, namely,

$$U_u(f, \hat{s}, s) = \sum_{t=1}^{T} u(f(\hat{s}_t), s_t),$$

where $s = (s_1, \ldots, s_T)$ are the ground truth states and $\hat{s} = (\hat{s}_1, \ldots, \hat{s}_T)$ are the predicted states. Note that due to our assumption that the decision maker's utility $u$ is linear and Lipschitz in its second argument, the policy evaluation function $U_u$ will be linear and Lipschitz in its last (ground-truth) argument $s$.

Our main results will all follow from appropriately instantiating our event collection such that it guarantees the following property of our sequence of predicted states, relative to (relevant) prediction-to-action policies.

**Definition C.2** (Transparent Policy Evaluation). Fix a prediction-to-action policy $f : \mathcal{S} \to \mathcal{A}$ and a policy evaluation function $U : (\mathcal{S} \to \mathcal{A}) \times \mathcal{S}^* \times \mathcal{S}^* \to \mathbb{R}$. Consider any two sequences of states $s, \hat{s} \in \mathcal{S}^*$ with $\text{len}(s) = \text{len}(\hat{s}) < \infty$. We interpret $s$ as the ground truth state sequence, and $\hat{s}$ as a predicted state sequence.

Then, the predictions $\hat{s}$ are $(f, U, s)$-*transparent* if:

$$U(f, \hat{s}, s) = U(f, \hat{s}, \hat{s}).$$

Henceforth, we will keep the ground truth sequence $s$ implicit and refer to predictions $\hat{s}$ as $(f, U)$-*transparent*.

This notion of transparency it at the core of our proposal for how to define trustworthy predictions in decision pipelines: If the predictions are $(f, U_u)$-transparent, then a decision maker with utility $u$ who employs prediction-to-action policy $f$ can safely view the predictions $\hat{s}$ as exactly coinciding with the ground truth states $s$ — in the sense that she can measure the performance of $f$ (using $U$) against the predictions rather than against the true states. This is often a useful property on its own — but when it holds for multiple prediction-to-action policies $f$, then optimizing amongst these policies on the basis of the predicted outcomes implies performance guarantees corresponding to optimality within this benchmark set.

**Enforcing Transparency via (Full) Calibration** We now see how full calibration lends transparency to *all* prediction-to-action policies with respect to *all* evaluation functions $U_u$ as defined above.

**Theorem C.3** (Calibration Lends Transparency to All Prediction-to-Action Policies). *Fix any time horizon $T$. Consider any ground truth sequence of states $s = (s_1, \dots, s_T)$ and any fully calibrated sequence of predictions $\hat{s} = (\hat{s}_1, \dots, \hat{s}_T)$, meaning that for all $v \in \{s_1, \dots, s_T\}$, it holds that $\sum_{t \in [T] : \hat{s}_t = v} s_t = v \cdot \#\{t \in [T] : \hat{s}_t = v\}$. Then, $\hat{s}$ is $(f, U_u)$-transparent for all $f : \mathcal{S} \to \mathcal{A}$ and every $u : \mathcal{A} \times \mathcal{S} \to \mathbb{R}$ that is linear in its second argument.*

*Proof.* The proof follows directly by definition of (full) calibration (equality (2)) and by linearity of $u$ in the state (equalities (1) and (3)). Letting $n_v := \#\{t \in [T] : \hat{s}_t = v\}$ for any value $v \in \mathbb{R}$, we get:

$$U_u(f, \hat{s}, s) = \sum_{t=1}^{T} u(f(\hat{s}_t), s_t) = \sum_{v \in \mathbb{R}} \sum_{t \in [T] : \hat{s}_t = v} u(f(v), s_t) \overset{(1)}{=} \sum_{v \in \mathbb{R}} u \left( f(v), \sum_{t \in [T] : \hat{s}_t = v} s_t \right)$$

$$\overset{(2)}{=} \sum_{v \in \mathbb{R}} u\left( f(v), v \cdot n_v \right) \overset{(3)}{=} \sum_{v \in \mathbb{R}} n_v \cdot u(f(v), v) = \sum_{t=1}^{T} u(f(\hat{s}_t), \hat{s}_t) = U_u(f, \hat{s}, \hat{s}). \qquad \square$$

**Using Transparency for Downstream Optimization**   Transparency on its own is valuable insofar as it means that a decision maker can follow a prediction-to-action policy $f$ with respect to predicted outcomes without being surprised about her long-run utility. But as we will now observe, it is also directly useful to decision makers for the purposes of optimizing their prediction-to-action policy. In fact, we will now see that enforcing the transparency property over all policies $f$ in any prediction-to-action policy class $\mathcal{F}$ that includes the best-response policy $f_u^{\mathrm{BR}}(\cdot) := \mathrm{BR}_u(\cdot)$ lets the decision maker obtain no regret to any other policy in that class $\mathcal{F}$ by simply playing $f_u^{\mathrm{BR}}$ in all rounds (i.e., trusting the predictions and acting accordingly).

**Theorem C.4** (Transparency over Policy Class $\mathcal{F}$ Implies Best-Response Optimality over $\mathcal{F}$). *Consider a decision maker with utility function $u : \mathcal{A} \times \mathcal{S} \to \mathbb{R}$. Consider any collection of prediction-to-action policies $\mathcal{F} \subseteq \mathcal{S}^{\mathcal{A}}$ such that the decision maker's best response policy is included in it: $f_u^{\mathrm{BR}} \in \mathcal{F}$.*

*Suppose the sequence of predictions $\hat{s}$ is $(f, U_u)$-transparent for all $f \in \mathcal{F}$. Then, committing to $f_u^{\mathrm{BR}}$ gives the decision maker no regret with respect to the policy class $\mathcal{F}$:*

$$U_u(f_u^{\mathrm{BR}}, \hat{s}, s) = \max_{f \in \mathcal{F}} U_u(f, \hat{s}, s).$$

*Proof.* Fix any policy $f \in \mathcal{F}$. By the definition of transparency (noting that $f_u^{\mathrm{BR}} \in \mathcal{F}$) and the definition of the best response policy:

$$U_u(f_u^{\mathrm{BR}}, \hat{s}, s) - U_u(f, \hat{s}, s) = U_u(f_u^{\mathrm{BR}}, \hat{s}, \hat{s}) - U_u(f, \hat{s}, \hat{s}) = \max_{f' : \mathcal{S} \to \mathcal{A}} U_u(f', \hat{s}, \hat{s}) - U_u(f, \hat{s}, \hat{s}) \geq 0,$$

implying the desired statement. $\qquad \square$

As an immediate corollary, we see that full calibration implies that the best response policy is simultaneously optimal for all downstream decision makers, amongst all prediction-to-action policies.

**Corollary C.5** (Calibration Implies Global Optimality of Best-Response Policy to All Decision-Makers). *Suppose the predictions $\hat{s}$ are fully calibrated. Then, simultaneously for all downstream decision makers (i.e., for all utilities $u : \mathcal{A} \times \mathcal{S} \to \mathbb{R}$), playing the best-response policy $f_u^{\mathrm{BR}}$ gives the decision maker no regret to all prediction-to-action policies:*

$$U_u(f_u^{\mathrm{BR}}, \hat{s}, s) = \max_{f : \mathcal{S} \to \mathcal{A}} U_u(f, \hat{s}, s) \quad \text{for all decision makers' utilities } u : \mathcal{A} \times \mathcal{S} \to \mathbb{R}.$$

*Proof.* As established, full calibration gives transparency to every decision maker (with any utility $u$) with respect to the entire class of all prediction-to-action policies $\mathcal{F}_{\mathrm{full}} := \mathcal{S}^{\mathcal{A}}$. Thus, by the preceding theorem, playing the best-response policy gives no regret to *all* $f : \mathcal{S} \to \mathcal{A}$. $\qquad \square$

## C.2. Transparency via Level-Set Unbiasedness and Swap Regret

Let us re-examine what properties our predictors should have in order to achieve transparency for various prediction-to-action policies. Fix any such policy $f : \mathcal{S} \to \mathcal{A}$, and suppose for a moment that we only need $(f, U_u)$ transparency for this specific $f$ and for all $u : \mathcal{A} \times \mathcal{S} \to \mathbb{R}$.

We already know that calibration — which demands unbiasedness from our predictions conditional on every possible prediction value $v \in \mathcal{S}$ — is sufficient for this purpose (Foster & Vohra, 1999). But it is not necessary. Intuitively, this is because the total utility $U_u$ of policy $f$ does not require such granular predictions for estimation. In particular, consider the collection of level sets of policy $f$, defined as $\mathrm{LS}(f) := \{f^{-1}(a)\}_{a \in \mathcal{A}}$. These level sets form a partition of the state space $\mathcal{S}$, but unless the mapping $f : \mathcal{S} \to \mathcal{A}$ is injective (which would necessitate the action space $\mathcal{A}$ being at least as complex as the space of predictions $\mathcal{S}$), this partition will be (likely much) less granular than the partition of $\mathcal{S}$ into single points as required by full calibration. To determine which action to play, policy $f$ only requires knowledge of the level set that the prediction belongs to, not the exact predicted value — and in this sense, $\mathrm{LS}(f)$ provides the right level of granularity over the state space $\mathcal{S}$ for us to confidently estimate the total utility of $f$. We formalize this as follows.

**Theorem C.6** (Transparent Policy Evaluation via Level-Set Unbiasedness). *Consider any policy $f : \mathcal{S} \to \mathcal{A}$. Suppose the predictions $\hat{s}$ are unbiased on the level sets of $f$, in the sense that for each level set $V \in \mathrm{LS}(f)$ (note that $V \subseteq \mathcal{S}$) it holds that $\sum_{t \in [T] : \hat{s}_t \in V} \hat{s}_t - s_t = 0$. Then, the predictions $\hat{s}$ are $(f, U_u)$-transparent for all possible decision makers' utilities $u : \mathcal{A} \times \mathcal{S} \to \mathbb{R}$.*

*Proof.* For each level set $V \in \mathrm{LS}(f)$, let $f(V) \in \mathcal{A}$ denote the action to which $f$ maps every prediction in $V$.

$$U_u(f, \hat{s}, s) = \sum_{t=1}^{T} u(f(\hat{s}_t), s_t) = \sum_{V \in \mathrm{LS}_f} \sum_{t \in [T] : \hat{s}_t \in V} u(f(\hat{s}_t), s_t) = \sum_{V \in \mathrm{LS}_f} \sum_{t \in [T] : \hat{s}_t \in V} u(f(V), s_t)$$

$$= \sum_{V \in \mathrm{LS}_f} u\left(f(V), \sum_{t \in [T] : \hat{s}_t \in V} s_t\right) = \sum_{V \in \mathrm{LS}_f} u\left(f(V), \sum_{t \in [T] : \hat{s}_t \in V} \hat{s}_t\right) = \sum_{V \in \mathrm{LS}_f} \sum_{t \in [T] : \hat{s}_t \in V} u(f(V), \hat{s}_t)$$

$$= \sum_{V \in \mathrm{LS}_f} \sum_{t \in [T] : \hat{s}_t \in V} u(f(\hat{s}_t), \hat{s}_t) = \sum_{t=1}^{T} u(f(\hat{s}_t), \hat{s}_t) = U_u(f, \hat{s}, \hat{s}). \qquad \square$$

But in fact, this result can be significantly strengthened at no cost. Consider the set $\Phi_{\mathcal{A}} = \{\phi : \mathcal{A} \to \mathcal{A}\}$ of all self-maps of the action set $\mathcal{A}$. For reasons that will become clear very soon, we will also refer to a self-map $\phi \in \Phi_{\mathcal{A}}$ as a *swap*. As it turns out, predictions that are unbiased on the level sets $\mathrm{LS}(f)$ of a policy $f$ lend transparency not just to the $f$ itself but also to each prediction-to-action policy $f_{\phi}$ that is a post-processing of the map $f$ by a swap $\phi \in \mathcal{A}$, that is, $f_{\phi} = \phi \circ f$.

**Theorem C.7** (Level-Set Unbiasedness Gives Transparency under All Swaps). *Consider any policy $f : \mathcal{S} \to \mathcal{A}$. As in the above theorem, suppose the state predictions $\hat{s}$ are unbiased on all level sets $V \in \mathrm{LS}(f)$ of $f$. Then, they are $(\phi \circ f, U_u)$-transparent for all swaps $\phi : \mathcal{A} \to \mathcal{A}$ and for all decision makers' utilities $u : \mathcal{A} \times \mathcal{S} \to \mathbb{R}$.*

*Proof.* Fix any swap $\phi : \mathcal{A} \to \mathcal{A}$. Then, the level sets of $\phi \circ f$ either coincide with, or are strictly coarser than, the level sets of $f$. Indeed, viewing $\mathrm{LS}(f)$ and $\mathrm{LS}(\phi \circ f)$ as partitions of $\mathcal{S}$, it is easy to see that $\mathrm{LS}(f)$ is a *refinement* of $\mathrm{LS}(\phi \circ f)$, in the sense that for any $V \in \mathrm{LS}(f)$ there exists some $V' \in \mathrm{LS}(\phi \circ f)$ with $V \subseteq V'$. As a result, each level set of $\phi \circ f$ is a disjoint union of one or more level sets of $f$. Thus, since the predictions $\hat{s}$ are unbiased on the level sets $\mathrm{LS}(f)$ of $f$, they are also unbiased on the level sets $\mathrm{LS}(\phi \circ f)$ of $\phi \circ f$, implying by the above theorem that they are $(\phi \circ f, U_u)$-transparent for all decision maker's utilities $u$. $\qquad \square$

This strengthened result is very useful because it implies *no swap regret guarantees* for decision makers when the predictions $\hat{s}$ are unbiased on the *level sets of the decision maker's best-response policy*.

**Theorem C.8** (No Swap Regret via Unbiasedness on Best-Response Level Sets). *Fix any decision maker with utility function $u : \mathcal{A} \times \mathcal{S} \to \mathbb{R}$. Consider her best-response policy $f_u^{\mathrm{BR}} : \mathcal{S} \to \mathcal{A}$. Then, if the predictions $\hat{s}$ are unbiased on the level sets $\mathrm{LS}(f_u^{\mathrm{BR}})$ of the best-response policy, the decision maker will obtain* no swap regret *by employing the best-response policy:*

$$U_u\left(f_u^{\mathrm{BR}}, \hat{s}, s\right) = \max_{\phi : \mathcal{A} \to \mathcal{A}} U_u\left(\phi \circ f_u^{\mathrm{BR}}, \hat{s}, s\right).$$

*Proof.* Since the predictions $\hat{s}$ are $(\phi \circ f_u^{\mathrm{BR}}, U_u)$-transparent for all swaps $\phi : \mathcal{A} \to \mathcal{A}$, by the definition of the best-response policy, we get:

$$U_u\left(f_u^{\mathrm{BR}}, \hat{s}, s\right) - \max_{\phi:\mathcal{A}\to\mathcal{A}} U_u\left(\phi \circ f_u^{\mathrm{BR}}, \hat{s}, s\right) = U_u\left(f_u^{\mathrm{BR}}, \hat{s}, \hat{s}\right) - \max_{\phi:\mathcal{A}\to\mathcal{A}} U_u\left(\phi \circ f_u^{\mathrm{BR}}, \hat{s}, \hat{s}\right) = 0. \qquad \square$$

## D. Faster Unbiased Prediction for Disjoint Events

In this section we show how to find an $\epsilon$-approximate solution to the minimax problems $\min \max u_t$ at all rounds $t \in [T]$, defined in Section 2, with running time that is polynomial in $d$, $|\mathcal{E}|$, and $\log(1/\epsilon)$ in the case in which the events $E \in \mathcal{E}$ are binary valued and disjoint: for all $x, \hat{s}$: $\sum_{E \in \mathcal{E}} E(x, \hat{s}) \leq 1$. We will also assume that for every history $\pi$ and context $x$, the predictions $\hat{s}$ that satisfy $E(\pi, x, \hat{s}) = 1$ form a convex set for which we have a polynomial time separation oracle.

**Throughout this appendix, we refer to weights $w_t$ from Section 2 as $q_t$, and to the randomized strategies $\bar{s}_t$ from Section 2 as $\psi_t$.**

Our goal is to solve for the learner's equilibrium strategy $\psi_t \in \Delta(\mathcal{S})$ in the game with utility function

$$u_t(\hat{s}, s) = \sum_{i=1}^{d} \sum_{\sigma \in \{-1,1\}} \sum_{E \in \mathcal{E}} q_{t,(i,\sigma,E)} \cdot \sigma \cdot E(\pi_{t-1}, x_t, \hat{s}) \cdot (\hat{s}_i - s_i)$$

corresponding to the per-round gain of MsMwC. In other words, we need to approximately solve:

$$\psi_t^* = \operatorname*{argmin}_{\psi \in \Delta(\mathcal{S})} \max_{s \in \mathcal{S}} \mathbb{E}_{\hat{s} \sim \psi} [u_t(\hat{s}, s)]. \tag{1}$$

By relaxing the minimization player's domain from $\mathcal{S}$ to $\Delta(\mathcal{S})$, the set of distributions over predictions, we have made the objective linear (and hence convex/concave), but we have continuously many optimization variables — both primal variables (for the minimization player) and dual variables (for the maximization player). Our strategy for solving this problem in polynomial time will be to argue that it has a solution in which only $|\mathcal{E}|$ many primal variables take non-zero values, that we can efficiently identify those variables, and that we can implement a separation oracle for the dual "constraints" in polynomial time. This will allow us to construct a reduced but equivalent linear program that we can efficiently solve with the Ellipsoid algorithm.

We first observe that in the utility function $u_t(\hat{s}, s)$, the learner's predictions $\hat{s}$ "interact" with the outcomes $s$ only through the activation of the events $E(\pi_{t-1}, x_t, \hat{s})$. This implies that *conditional* on the values of the events $E(\pi_{t-1}, x_t, \hat{s})$, there is a unique $\hat{s}$ that minimizes $u_t(\cdot, s)$ *simultaneously for all $s$*. In general, the collection of events $E(\pi_{t-1}, x_t, \hat{s})$ could take on many different combinations of values — but our assumption in this section that the events are disjoint and binary means that there are in fact only $|\mathcal{E}|$ different candidate values of $\hat{s}$ for us to consider — namely, those defined by the following efficiently solvable convex programs:

**Definition D.1.** For $E \in \mathcal{E}$, let $\hat{s}_t^{*,E}$ be a solution to the following convex program (selecting arbitrarily if there are multiple optimal solutions):

$$\operatorname{minimize}_{\hat{s} \in \mathcal{S}} \quad \sum_{i=1}^{d} \sum_{\sigma \in \{-1,1\}} q_{t,(i,\sigma,E)} \cdot \sigma \cdot \hat{s}_i$$

$$\text{subject to} \quad E(\pi_{t-1}, x_t, \hat{s}) = 1.$$

Let $\mathcal{P}_t = \{\hat{s}_t^{*,E}\}_{E \in \mathcal{E}}$ be a collection of $|\mathcal{E}|$ vectors in $\mathcal{S}$ constituting solutions to the above programs.

*Remark* D.2. As we have assumed in this section, the set of $\hat{s}$ such that $E(\pi_{t-1}, x_t, \hat{s}) = 1$ is a convex region endowed with a separation oracle, and so these are indeed convex programs that we can efficiently solve with the Ellipsoid algorithm. This is often the case: for example, if we have a decision maker with a utility function $u$ over $K$ actions, the disjoint binary events $E_{u,a}$ (for each action $a \in [K]$) are defined by $K$ linear inequalities, and so form a convex polytope with a small number of explicitly defined constraints; this collection of events is relevant for obtaining diminishing swap regret for downstream decision makers.

We next verify that the prediction values defined in Definition D.1 are best responses for the minimization player against all possible realizations $s_t$ that the maximization player might choose, *conditional* on a positive value of a particular event:

**Lemma D.3.** *Simultaneously for all $s \in \mathcal{S}$, we have:*

$$\hat{s}_t^{*,E} \in \underset{\hat{s}:E(\pi_{t-1},x_t,\hat{s})=1}{\arg\min} u_t(\hat{s}, s).$$

A consequence of this is that solutions to the following reduced minimax problem (which now has only $|\mathcal{E}|$ variables for the minimization player — the weights defining a distribution over the $|\mathcal{E}|$ points $\hat{s}_t^{*,E}$ ) are also solutions to our original minimax problem 1:

$$\psi_t^* = \underset{\psi \in \Delta(\mathcal{P}_t)}{\arg\min} \max_{s \in \mathcal{S}} \underset{\hat{s} \sim \psi}{\mathbb{E}} [u_t(\hat{s}, s)]. \tag{2}$$

**Lemma D.4.** *Fix any optimal solution $\psi_t^*$ to minimax problem 2. Then $\psi_t^*$ is also an optimal solution to minimax problem 1.*

Thus, to find a solution to minimax problem 1, it suffices to find a solution to minimax problem 2. Minimax problem 2 can be expressed as a linear program with $|\mathcal{E}| + 1$ variables but with continuously many constraints, one for each $s \in \mathcal{S}$:

$$\begin{aligned}
&\text{minimize}_{\psi \in \Delta(\mathcal{P}_t)} \quad \gamma \\
&\text{subject to} \\
&\qquad\qquad \mathbb{E}_{\hat{s} \sim \psi}[u_t(\hat{s}, s)] \leq \gamma \quad \forall s \in \mathcal{S}.
\end{aligned} \tag{3}$$

We can find an $\epsilon$-approximate solution to a polynomial-variable linear program using the Ellipsoid algorithm in time polynomial in the number of variables and $\log(1/\epsilon)$ so long as we have an efficient *separation oracle* — i.e., an algorithm to find an $\epsilon$-violated constraint whenever one exists, given a candidate solution. In this case, implementing a separation oracle corresponds to computing a *best response* for the adversary (the maximization player) in our game—and since the utility function in our game is *linear* in the adversary's chosen action $s$, implementing a separation oracle corresponds to solving a linear maximization problem over the convex feasible region $\mathcal{S}$—a problem that we can solve efficiently assuming we have a separation oracle for $\mathcal{S}$. There are a number of technical details involved in making this rigorous, which can be found in Appendix D. Here we state the final algorithm and guarantee.

---

**Algorithm 3** `Get-Approx-Equilibrium-LP` $(t, \epsilon, \mathcal{E})$

---

   **for** $E \in \mathcal{E}$ **do**
      Solve the convex program from Definition D.1 to obtain $\hat{s}_t^{*,E}$.
   **end for**
   Let $\mathcal{P}_t = \{\hat{s}_t^{*,E}\}_{E \in \mathcal{E}}$.
   Solve linear program 3 over $\mathcal{P}_t$ using the weak Ellipsoid algorithm to obtain solution $\psi_t'$.
   Let $\psi_t^*$ be the Euclidean projection of $\psi_t'$ onto $\Delta(\mathcal{P}_t)$ returned by the simplex projection algorithm.
   Return $\psi_t^*$.

---

**Theorem D.5.** *Given a polynomial-time separation oracle for $\mathcal{S}$, for any $\epsilon > 0$, there exists an algorithm (Algorithm 3) that returns an $\epsilon$-approximately optimal solution $\psi_t^*$ to minimax problem 1 and runs in time polynomial in $d$, $|\mathcal{E}|$, $\log(\frac{1}{\epsilon})$.*

We solve for an $\epsilon$-approximate solution of linear program 3 using a *weak separation oracle*, using an approximate version of the Ellipsoid algorithm.

**Definition D.6.** For any $\epsilon > 0$ and any convex set $S$, let

$$S^{+\epsilon} = \{s : ||s - \tilde{s}||_2 \leq \epsilon \quad \text{for some } \tilde{s} \in S\} \quad S^{-\epsilon} = \{s : B_2(s, \epsilon) \subseteq S\}$$

be the positive and negative $\epsilon$-approximate sets of $S$, where $B_2(x, r)$ is a ball of radius $r$ under the $\ell_2$ norm.

**Definition D.7.** A *weak separation oracle* for a convex set $S$ is an algorithm that, when given input $\psi \in \mathbb{Q}^d$ and positive $\epsilon \in \mathbb{Q}$, confirms that $\psi \in S^{+\epsilon}$ if true, and otherwise returns a hyperplane $a \in \mathbb{Q}^d$ such that $||a||_\infty = 1$ and $\langle a, \psi \rangle \leq \langle a, \psi' \rangle + \epsilon$ for all $\psi' \in S^{-\epsilon}$.

We express a separation oracle for linear program 3 as the convex program that solves for the most violated constraint given a candidate solution $\psi$, which is simply the best response problem for the maximization player in minimax problem 2. This is the problem of maximizing a $d$-variable linear function over the convex set $\mathcal{S}$. To make sure that we can control the bit

complexity of the constraint returned by the separation oracle we round the coordinates of the constraint $a \in \mathbb{R}^d$ output by the separation oracle to a rational-valued vector within $\pm \frac{\epsilon}{2}$ of the exact solution by truncating each coordinate of $a$ to $\log(\frac{1}{\epsilon})$ bits.

**Definition D.8.** A solution $\psi \in S^{+\epsilon}$ is $\epsilon$-*weakly optimal* if, given $\epsilon > 0$, $\mathbb{E}_{\hat{s} \sim \psi}[u(\hat{s}, s)] \leq \mathbb{E}_{\hat{s} \sim \psi'}[u(\hat{s}, s)] + \epsilon$ for all $\psi' \in S^{-\epsilon}$ and for all $s$.

For an $\epsilon$-approximate solution to minimax problem 2, it suffices to find an $\epsilon$-weakly optimal solution to linear program 3, which we can do using the Ellipsoid method. However, the solution to the weak optimization may not even be a valid probability distribution (since it only approximately satisfies the constraints) – in this case, we can project our infeasible solution back to feasibility. We use the simplex Euclidean projection algorithm given by (Condat, 2016) to project the candidate solution back to a feasible region and show that this projected feasible solution is still $\epsilon$-approximately optimal.

**Theorem D.5.** *Given a polynomial-time separation oracle for $\mathcal{S}$, for any $\epsilon > 0$, there exists an algorithm (Algorithm 3) that returns an $\epsilon$-approximately optimal solution $\psi_t^*$ to minimax problem 1 and runs in time polynomial in $d$, $|\mathcal{E}|$, $\log(\frac{1}{\epsilon})$.*

*Proof.* Linear program 3 encodes minimax problem 2. To solve LP 3, we use the Ellipsoid algorithm, which gives an approximate solution in polynomial time under the following conditions:

**Theorem D.9** ((Grötschel et al., 1988), Theorem 4.4.7). *Given a weak separation oracle over convex constraint set $S$ and $\epsilon > 0$, the Ellipsoid algorithm finds a $\epsilon$-weakly optimal solution over $S$ in time polynomial in the bit complexity of the constraints returned by the separation oracle, the bit complexity of the objective function, and the bit complexity of $\epsilon$.*

Fix some $\epsilon > 0$. Let $S$ be the constraint set, which are a set of linear constraints over a convex compact set (i.e. $s \in \mathcal{S}$) and constraints enforcing a probability simplex (i.e. $\psi \in \Delta(\mathcal{P})$), implying that $S$ is a convex set. Let $\epsilon' = \frac{\epsilon}{2C\sqrt{|\mathcal{E}|}}$. Given an exact separation oracle over $\mathcal{S}$, preserving $\log(\frac{1}{\epsilon'})$ bits of the most violated constraint given by the separation oracle and rounding to a rational number yields an rational $\epsilon'$-approximate most violated constraint, which satisfies the conditions for a weak separation oracle. Thus, we can find an $\epsilon'$-weakly optimal solution $(\gamma', \psi')$ to minimax problem 2, where $\psi' \in S^{+\epsilon}$. In the case that $\psi'$ is a valid probability distribution, we have found an $\epsilon$-approximate optimal solution $\psi_t^* = \psi'$.

Otherwise, $\psi'$ may violate conditions for a valid probability distribution if the linear constraints do not constrain the feasible set (i.e. $S = \Delta(P)$). Since $\psi' \in S^{+\epsilon}$, there exists some $\psi^\epsilon \in S$ such that $||\psi^\epsilon - \psi'|| \leq \epsilon'$. We find this point $\psi^\epsilon$ via the simplex projection algorithm in (Condat, 2016).

We show that this projection back to a feasible probability distribution still leaves us with an $\epsilon$-approximately optimal solution. Let $u_t(\hat{s}_t^*, s)$ be the $|\mathcal{E}|$-dimensional vector such that each coordinate $E$ has entry $u_t(\hat{s}_t^{*,E}, s)$. First, we show that $|u_t(\hat{s}_t^{*,E}, s)|$ is bounded by $C = 2 \max_{s \in \mathcal{S}} ||s||_\infty$ for $E \in \mathcal{E}$:

$$|u(\hat{s}_t^{*,E}, s)| \leq \sum_{i=1}^d \sum_{\sigma \in \{-1,1\}} \sum_{E \in \mathcal{E}} q_{t,(i,\sigma,E)} \cdot |\sigma| \cdot E(x_t, \hat{s}_t^{*,E}) \cdot |\hat{s}_{t,i}^{*,E} - s_i|$$

$$\leq \sum_{i=1}^d \sum_{\sigma \in \{-1,1\}} \sum_{E \in \mathcal{E}} q_{t,(i,\sigma,E)} \cdot |\hat{s}_{t,i}^{*,E} - s_i| \leq \sum_{i=1}^d \sum_{\sigma \in \{-1,1\}} \sum_{E \in \mathcal{E}} q_{t,(i,\sigma,E)} \cdot C = C,$$

where we used that $E(x_t, \hat{s}_t^{*,E}) \leq 1$ and $|\sigma| = 1$, and that $q \in \Delta(2d|\mathcal{E}|)$, implying it must sum to 1. From this, we find that $||u_t(\hat{s}_t^*, s)||_2 \leq \sqrt{C_1^2 + \ldots + C_{|\mathcal{E}|}^2} \leq C\sqrt{|\mathcal{E}|}$.

Next, by continuity of inner product, given $\epsilon > 0, s \in \mathcal{S}$, there exists $\delta > 0$ such that $||\psi^\epsilon - \psi'|| \leq \delta$ implies that $|| \mathbb{E}_{\hat{s} \sim \psi^\epsilon}[u_t(\hat{s}, s)] - \mathbb{E}_{\hat{s} \sim \psi'}[u_t(\hat{s}, s)]|| \leq \epsilon$. By Cauchy-Schwarz, we can bound the difference between the expectations as follows:

$$\left|\left| \mathbb{E}_{\hat{s} \sim \psi^\epsilon}[u_t(\hat{s}, s)] - \mathbb{E}_{\hat{s} \sim \psi'}[u_t(\hat{s}, s)] \right|\right|_2 = \langle \psi^\epsilon - \psi^*, u_t(\hat{s}_t^*, s) \rangle \leq ||\psi_t^\epsilon - \psi'||_2 \cdot ||u_t(\hat{s}_t^*, s)||_2 \leq \delta \cdot C\sqrt{|\mathcal{E}|}.$$

Thus, using $\psi_t^* = \psi^\epsilon$ as the solution and setting $\delta = \epsilon'$ gives us an $\epsilon' \cdot C\sqrt{|\mathcal{E}|} + \epsilon' = \frac{\epsilon}{2} + \frac{\epsilon}{2C\sqrt{|\mathcal{E}|}} \leq \epsilon$ approximate solution.

By Lemma D.4, any optimal solution to minimax problem 2 is an optimal solution to minimax problem 1, so we must have that $\psi_t^*$ is an $\epsilon$-approximate solution to minimax problem 1.

Now we consider the runtime of the algorithm. In order for LP 3 to be well-formulated, we first solve $|\mathcal{E}|$ convex programs (one for each $\hat{s}^{*,E}$), which takes time polynomial in $d$. Now, consider the bit complexity of the constraints. For the inequality constraints, the bit complexity of each constraint bounding the objective function is given by the bit complexity of $\mathbb{E}_{\hat{s} \sim \psi}[u(\hat{s}, s)]$. Each coefficient of $\psi(\hat{s}_t^{*,E})$ is $u_t(\hat{s}_t^{*,E}, s)$, which is bounded by $C$ from above. Since there are $|\mathcal{E}|$ variables in this constraint, the maximum bit complexity of any constraint is bounded by $O(\log(C|\mathcal{E}|))$. Similarly, the objective function has polynomial bit complexity on the scale of $O(\log(C|\mathcal{E}|))$. Finally, $\epsilon$ has a bit complexity of $\log(\frac{1}{\epsilon})$. The simplex projection algorithm has quadratic runtime in the dimension of the vector, which takes $O(|\mathcal{E}|^2)$ time. Thus, the runtime of the algorithm is polynomial in $d, |\mathcal{E}|, \log(C|\mathcal{E}|)$, and $\log(\frac{1}{\epsilon})$. $\qquad\square$

**Lemma D.3.** *Simultaneously for all $s \in \mathcal{S}$, we have:*

$$\hat{s}_t^{*,E} \in \underset{\hat{s}:E(\pi_{t-1}, x_t, \hat{s})=1}{\operatorname{argmin}} u_t(\hat{s}, s).$$

*Proof.* The constraint that $E(\pi_{t-1}, x_t, \hat{s}) = 1$ together with the fact that the set of events $\mathcal{E}$ is disjoint and binary implies that for all other events $E' \in \mathcal{E}$, $E'(\pi_{t-1}, x_t, \hat{s}) = 0$. For any $\hat{s}$ such that $E(\pi_{t-1}, x_t, \hat{s}) = 1$, we therefore have that $u_t(\hat{s}, s)$ reduces to:

$$u_t(\hat{s}, s) = \sum_{i=1}^{d} \sum_{\sigma \in \{-1, 1\}} q_{t,(i,\sigma,E)} \cdot \sigma \cdot \hat{s}_i - q_{t,(i,\sigma,E)} \cdot \sigma \cdot s_i.$$

But in this expression, the $\hat{s}$ terms have no interaction with the $s$ terms, and hence we have that for any $s$:

$$\underset{\hat{s}:E(\pi_{t-1}, x_t, \hat{s})=1}{\operatorname{argmin}} u_t(\hat{s}, s) = \underset{\hat{s}:E(\pi_{t-1}, x_t, \hat{s})=1}{\operatorname{argmin}} \left( \sum_{i=1}^{d} \sum_{\sigma \in \{-1, 1\}} q_{t,(i,\sigma,E)} \cdot \sigma \cdot \hat{s}_i \right) = \hat{s}_t^{*,E}. \qquad\square$$

**Lemma D.4.** *Fix any optimal solution $\psi_t^*$ to minimax problem 2. Then $\psi_t^*$ is also an optimal solution to minimax problem 1.*

*Proof.* We first observe that minimax problem 2 is only a more constrained problem for the minimization player than minimax problem 1, as $\mathcal{P}_t \subset \mathcal{S}$. Thus it suffices to show that given a solution $\hat{\psi}_t$ for minimax problem 1, we can transform it into a new solution $\psi_t$ such that:

1. $\psi_t$ has support only over points in $\mathcal{P}_t$, and

2. For all $s \in \mathcal{S}$, $\mathbb{E}_{\hat{s}_t \sim \psi_t}[u(\hat{s}_t, s)] \geq \mathbb{E}_{\hat{s}_t \sim \hat{\psi}_t}[u(\hat{s}_t, s)]$.

Given $\hat{\psi}_t$, we construct $\psi_t$ as follows: for each event $E$, we take all of the weight that $\hat{\psi}_t$ places on points $\hat{s}$ such that $E(\pi_{t-1}, x_t, \hat{s}) = 1$, and place that weight on $\hat{s}_t^{*,E} \in \mathcal{P}_t$:

$$\psi_t(\hat{s}_t^{*,E}) = \hat{\psi}_t(\{\hat{s} : E(\pi_{t-1}, x_t, \hat{s}) = 1\}).$$

By construction $\psi_t$ has support over points in $\mathcal{P}_t$. It remains to show that $\psi_t$ has objective value that is at least as high as $\hat{\psi}_t$ for every $s \in \mathcal{S}$:

$$
\begin{aligned}
\underset{\hat{s}_t \sim \hat{\psi}_t}{\mathbb{E}}[u(\hat{s}_t, s)] &= \sum_{E \in \mathcal{E}} \underset{\hat{s}_t \sim \hat{\psi}_t}{\operatorname{Pr}}[E(\pi_{t-1}, x_t, \hat{s}_t) = 1] \underset{\hat{s}_t \sim \hat{\psi}_t}{\mathbb{E}}[u(\hat{s}_t, s)|E(\pi_{t-1}, x_t, \hat{s}_t) = 1] \\
&\leq \sum_{E \in \mathcal{E}} \underset{\hat{s}_t \sim \hat{\psi}_t}{\operatorname{Pr}}[E(\pi_{t-1}, x_t, \hat{s}_t) = 1] u(\hat{s}_t^{*,E}, s) \\
&= \underset{\hat{s}_t \sim \psi_t}{\mathbb{E}}[u(\hat{s}_t, s)]
\end{aligned}
$$

The inequality follows from Lemma D.3. $\qquad\square$

