# OpenReview forum: "High-Dimensional Prediction for Sequential Decision Making"
_ICML.cc/2025/Conference — ICML 2025 oral_

### Official Review · Reviewer_RVzC · 2025-02-26

**Overall Recommendation:** 5

**Summary:**

This paper introduces a general framework for constructing sequential decision-making strategies by predicting the states of the environment. The basic idea is that if we can estimate the *future* states of the environment, we can make decisions based on these predictions. However, the challenge is that, since the states are generated adversarially, it is not possible to predict them accurately. Nevertheless, this paper demonstrates that for many sequential decision-making problems, it suffices to construct a predictor that is *unbiased* when weighted by certain events.

The main technical contribution is a general prediction strategy that achieves low weighted bias for arbitrary events of polynomial size. The construction is based on a clever reduction to the problem of prediction with expert advice. Furthermore, the paper applies the general state prediction strategy to several concrete problems, including online multicalibration, swap regret, and online combinatorial optimization.

I find the idea of reducing sequential decision-making to unbiased state estimation both interesting and original. I expect that this technique will be applicable to a broader range of problems and deserves wider recognition in the community. Therefore, I strongly recommend accepting this paper.

**Claims And Evidence:**

The claims are clear and supported by rigorous proofs, which are sound.

**Essential References Not Discussed:**

N/A

**Experimental Designs Or Analyses:**

N/A

**Methods And Evaluation Criteria:**

This is a pure theory paper. The authors apply their method to a broad range of problems, which is convincing.

**Other Comments Or Suggestions:**

- In line 259 (right), the definition of ECE must be $\sup_{p\in [0,1]}$, not $\sum_{p\in [0,1]}$.
- In page 5, you use both $w_{\sigma,i,j}^t$ and $w_{t,(\sigma,i,j)}$, I suggest sticking to $w_{\sigma,i,j}^t$.

**Other Strengths And Weaknesses:**

One complaint about the paper is that the notation is quite difficult to parse, e.g., the subscripts are messy. Moreover, it seems that the derived regret inevitably depends on the size of the decision space, which may not be applicable for large decision spaces. Although the authors provide an example in the context of online combinatorial optimization, it relies on a very specific additive property of the utility function.

**Questions For Authors:**

None.

**Relation To Broader Scientific Literature:**

This paper provides a general methodology for resolving sequential decision-making problems in adversarial environments, which unifies and simplifies many prior approaches.

**Theoretical Claims:**

I checked almost all the proofs in the main text and am convinced that they are correct. However, the paper also provides many additional results in the appendix, which I did not check and therefore cannot guarantee their correctness.

---

> ### Author Rebuttal · Authors · 2025-04-01
>
> We thank the reviewer for the insightful reading and positive assessment of our paper, and for the feedback! We will make sure to address the notational comments in our revision.

---

### Official Review · Reviewer_daBU · 2025-03-12

**Overall Recommendation:** 4

**Summary:**

This paper introduces a two-step framework for decision-making in an online adversarial setting in which a "master" algorithm makes predictions of vector-valued "states" that encode the decision-making relevant features of the environment by achieving low bias subject to a collection of conditioning events. Downstream agents may then consume those states through the a simple best-response strategy to enjoy strong regret guarantees themselves, such as online multicalibration or swap regret. The "master" algorithm is efficient so long as the number of conditioning events is polynomial or, in settings with combinatorially large action spaces, it has access to an optimization oracle. Crucially, the power of the master algorithm comes from properly defining a set of conditioning events, $\mathcal{E}$, which lead to the following downstream results in the paper.

The authors show three main applications of this "master" algorithm (and slight variants of it):
1. Efficient online multicalibration with $O(T^{2/3})$ rate in the expected calibration error metric.
2. A polynomial number of downstream agents can *all* achieve diminishing swap regret by just "best responding" to the "master" algorithm.
3. A polynomial number of downstream agents can *all* achieve diminishing "conditional" regret in a combinatorial optimization problem over a polynomial number of conditioning events *efficiently* given an optimization oracle for the combinatorial problem.

**Claims And Evidence:**

In my reading of the paper, I saw that there was one major claim in **Theorem 2.4**, which gives the bias guarantee of the "master" algorithm, Algorithm 1. The other claims in the paper follow from applying Theorem 2.4 in various ways by changing the events $\mathcal{E}$ for which Algorithm 1 ensures unbiased prediction against. In particular:

1. Section 2.2 shows an algorithm for achieving $O(T^{2/3})$ online (one-dimensional) multicalibration error by defining the events as the collection of groups cross the collection of discretizaiton of $[0, 1]$.
2. Theorem 3.7 demonstrates that $n$ downstream agents with $K$ actions can simultaneously achieve diminishing $O(\sqrt{KT})$ swap regret by best-responding to the "master algorithm" states, so long as the master algorithm is instantiated with the event set as a collection of $nK$ "best-response" events, or indicators that agent $i \in [n]$ had best response to the state equal to action $a$ (of the $K$ actions).
3. Theorem 4.1 demonstrates that $n$ downstream agents with combinatorial action sets granted oracle access to an offline optimization oracle playing best-response actions achieves $O(d \sqrt{T})$ where $d$ is the number of "base" actions in the action sets.

I checked the proofs and proof sketches that are present in the main body of the paper, and I believe they are correct.

**Essential References Not Discussed:**

To my knowledge, the authors discuss all the relevant work to the results of this paper.

**Experimental Designs Or Analyses:**

N/A -- this paper is mainly a theoretical work.

**Methods And Evaluation Criteria:**

The paper is a theoretical work in online learning, and the evaluation criteria of regret (specifically, notions such as online multicalibration and swap regret) are well established definitions in the literature. The proposed evaluation criteria make sense.

**Other Comments Or Suggestions:**

A couple of small suggestions:
- Page 4: I believe that the $w_{\sigma, i, j}$ should have a $t$ subscript as well.
- Page 4: I would suggest motivating the form of "Unbiased Prediction" in Definition 2.2 with the existing "small-loss" rates in case readers aren't familiar with these faster rates for regret. Or, to prevent confusion, it might be helpful to just denote "Unbiased Prediction" as decaying sublinearly, and then add a remark that the goal is to achieve the "fast rate" decay that MsMwC affords.

**Other Strengths And Weaknesses:**

**Strengths**
- The paper is well-written and the proofs of the main theorems are organized in an easy-to-understand manner.
- The "Predict-then-Act" framework the authors propose in its own right, and it presents a different model of looking at sequential decision-making that I believe is appealing and worth further investigation.
- The techniques for proving the main results are clean and appealing. I particularly enjoyed how the arguments take a similar form to the omniprediction/outcome indistinguishability literature, in which the "states" the master algorithm supplies are, in effect, indistinguishable from the true states so long as the downstream decision-makers are concerned with an appropriate collection of conditioning events.

**Weaknesses**
- I believe the main weakness of this work is that the "Predict-then-Act" motivation for these guarantees can be muddled in the exposition. One suggestion I have would be to begin the introduction with a motivating example to make clearer this framework, as readers may be more familiar with the online learning setting where there is a single agent making actions over $T$ rounds, possibly with access to a context. This additional "layer" of a "master" algorithm producing states for *multiple* agents to consume and then act on is interesting, but I found it difficult to follow at times without referring back to a guiding example.
- It would be helpful to present, in a more self-contained way, existing rates in the literature for the three applications you consider so the reader can disentangle and better understand the optimality of the results you propose. Perhaps a table, even in the Appendix, comparing the results to existing results in online multicalibration, swap regret, and online combinatorial optimization would be helpful.

**Questions For Authors:**

I had one main question for the authors concerning the multicalibration result:
1. Perhaps this is from my lack of familiarity with existing rates for multicalibration, but does the $O(T^{2/3})$ rate come from the fact that the algorithm you propose enjoys the "fast-rate" guarantee that scales with $n_T$ which allows a sharper tuning? In the first paragraph in Section 2.2, you claim that the fast $O(\sqrt{n_T})$ rates are useful, and I was wondering why exactly. I assumed that it was because existing methods for online multicalibration don't exploit such "small-loss" regret guarantees. This might also be worth clarifying in the main paper (second bullet point of "Weaknesses").

**Relation To Broader Scientific Literature:**

The key contributions of the paper are situated, in my view, in the literature in online learning, sequential decision-making, and calibration. Specifically, it follows a line of work started in [BM07] for guaranteeing online adversarial learning guarantees against a collection of *conditioning events*, originally termed "time selection functions." This original work motivates the authors' focus on swap regret, a very strong regret guarantee that has further implications in algorithmic game theory and correlated equilibria. The authors also design an algorithm for the online adversarial version of multicalibration, a desideratum originally introduced for the algorithmic fairness literature by [HKRR18] (albeit in a batch/i.i.d. formulation). Finally, the work touches on the literature on optimization in combinatorially large action spaces, where the Learner is assumed to have access to an offline optimization oracle. These "oracle-efficient" regret guarantees follow the early work of [KV05].

[BM07] Blum, Avrim, and Yishay Mansour. "From external to internal regret." Journal of Machine Learning Research 8.6 (2007).
[HKRR18] Hébert-Johnson, U., Kim, M., Reingold, O., & Rothblum, G. (2018, July). Multicalibration: Calibration for the (computationally-identifiable) masses. In International Conference on Machine Learning (pp. 1939-1948). PMLR.
[KV05] Kalai, A., & Vempala, S. (2005). Efficient algorithms for online decision problems. Journal of Computer and System Sciences, 71(3), 291-307.

**Theoretical Claims:**

I checked all the proofs that were included in the main body, and they were correct in my understanding.

---

> ### Author Rebuttal · Authors · 2025-04-01
>
> We thank the reviewer for the careful reading and positive assessment of our paper, and appreciate the insightful comments! We agree with, and will make sure to address, your comments in the revision. Among other things, we completely agree that starting with a single agent case would further enhance the exposition and readability; in fact, the only reason that prevented us from doing so was the page limit on the submission. Also, having a condensed reference paragraph/table for prior state of the art in the directions of our applications will help further clarify the landscape for the readers. Furthermore, we completely agree that first defining unbiased prediction as requiring sublinear (o(T)) bias, and only then instantiating it with the concrete regret bounds, is an excellent expository idea --- and in fact, we had our presentation laid out just like that in the pre-submission manuscript, and had to cut it down to the direct small-loss definition purely because of the page limit.
>
> We now give a clarification regarding your question about the utility of sharp “small loss”, group-specific, rates. In the multi-calibration example, please note that for each group, as stated on line 266, the bound is O(T/m + \sum_{i \in [m]} \sqrt{\text{# rounds on which bucket i got played}}). As mentioned further in the argument, by concavity of the sqrt function, we can conclude that the worst-case split of the T rounds into the m buckets is when each bucket gets played equally often, i.e. the worst case bound is $O(T/m + m \sqrt{T/m}) = O(T/m + \sqrt{m T})$, which when choosing the optimal m (that equalizes both terms), results in $m \sim T^{1/3}$ and thus in the overall per-group bound of $O(T^{2/3})$.
>
> Meanwhile, suppose our framework only gave $\sqrt{T}$ bounds for each group rather than \sqrt{E[\text{# rounds on which each group appeared}]}. Then, the calibration error bound for each group would simply be $O(T/m + \sum_{i \in [m]} \sqrt{T}) = O(T/m + m \sqrt{T})$, which due to an extra $\sqrt{m}$ factor in the right-hand term leads to the choice of $m \sim T^{1/4}$ and hence implies a worse $O(T^{3/4})$ error bound. In this way, our ability to use our framework without any modifications to get the (currently best-known) rate very concretely hinges on the “small-loss” per-group guarantees. And indeed, as you correctly assumed, the prior method of Gupta et al (2021) didn’t achieve such small-loss bounds and could thus only achieve the 3/4 exponent.

---

### Official Review · Reviewer_MMHe · 2025-03-15

**Overall Recommendation:** 4

**Summary:**

The paper advances sequential prediction in high-dimensional settings while maintaining unbiasedness across a possible set of events. For this, the paper utilises the multipliscale multiplicative weights with correction algorithm and the standard minmax machinery that is common in this line of work. This unbiased across a set of events prediction is then used to derive a range of other properties: like swap-regret control for downstream decision makers who will best respond to such predictions, online multi-calibration with favorable rates, and control over conditional regret in online optimisation problem.

**Claims And Evidence:**

The paper makes convincing claims with proper evidence.

**Essential References Not Discussed:**

not applicable

**Experimental Designs Or Analyses:**

Not applicable.

**Methods And Evaluation Criteria:**

The paper is primarily theoretical, and is rigorous in that sense.

**Other Comments Or Suggestions:**

None

**Other Strengths And Weaknesses:**

The paper is generally well-written as is easier to follow despite being dense. As stated above, the proposed approach is general (however there are costs to generality, and I'd appreciate some discussion around that).

**Questions For Authors:**

Some questions that can get some clarifications are:
1. What are the practical limits of specifying the finite collection of events E. Are there heuristic or data-driven methods for identifying the most relevant events, especially in high-dimensional contexts? Is it feasible to adaptively add or remove events over time? I assume one can plug-in new events sequentially, or my understanding wrong?

2. Related to above: guaranteeing no-swap-regret, as proposed, is based on explicitly constructing ‘best-response events’ for each agent’s utility, and knowing there are exactly n decision-makers. However, practically one might not know an agent’s utility function in advance, or there is only a partial information, plus new agents with different utilities may arrive at any time step. Could the authors discuss whether their approach can be adapted to handle unknown or changing utilities? In particular, is there a way to achieve similar no-swap-regret guarantees without having to explicitly compute best-response events for each agent ahead of time? I assume the traditional calibration (binary forecasts) result in swap-regret guarantees without any specification of the utility function?

3. The paper treats each round's state $s_t$ as "sufficient statistics"  for
downstream decision-making, implicitly assuming that an agent's utility depends only on
these $d$ coordinates. What is the nature of the sufficient statistics: is it just high-dimensional object or encapsulates higher order moments like variance etc.?  I assume not, as the decision-maker still has the linear utility in terms of the state. So I guess I'm asking what does "sufficient statistics" mean in the context of the paper, as it states sufficient for all decision-makers?

**Relation To Broader Scientific Literature:**

The paper is certainly relevant on multiple fronts. While the paper does not propose significant technical tools, Algroithm 1 is based on linearising the bias objective, and then to use the standard argument of min-max theorem to control the bias, the work itself is consequential as in encapsulates a range of related yet distinct problems: for example multi-calibration in online setting with rates of $T^{2/3}$, decision-making for any polynomial number of agents---achieving such decision-making guarantee algorithmically is challenging, and conditional regret guarantees. This work unifies multiple lines of research---calibration, no-regret learning, and combinatorial optimisation---into one efficient framework that handles arbitrary polynomial-size event families.
Overall, this is a solid paper.

**Theoretical Claims:**

I haven't checked the written proofs word by word, but I do get the intuition and the argument of the proofs.

---

> ### Author Rebuttal · Authors · 2025-04-01
>
> We thank the reviewer for the careful reading and positive assessment of our paper, and for the very insightful feedback! Indeed, as pointed out in your review, our framework is quite general, and discussing the costs of this generality is very pertinent and we will do so in the revision. To address specific points you have raised:
>
> 1. Regarding how to specify, and work with, the event collection. Indeed, our framework allows you to flexibly add new events to the event collection at any time point, without advance notice to the algorithm, as well as — conversely — terminate events that are not useful anymore. All this while barely incurring extra cost (cf. log |E| in the bias bound) for doing so. This is enabled by the fact that our algorithm at each round only works with events that are currently active and doesn’t require knowledge about other events, whether or not they were active before or will be active later.
> In terms of data-driven approaches, indeed in the presence of high-dimensional covariates one can for instance take a dimensionality-reducing bounded representation map, which, when applied to the covariates, results in much smaller-length vectors e.g. in the [0, 1]^d-hypercube, and define events e.g. as the outputs (or a more complex function) of this representation. E.g. the case of linear, bounded, representations give rise to the “multiaccurate” or “multigroup” setting in the literature on algorithmic fairness, and the groups there can be defined either normatively as e.g. demographic groups (age/income/etc), or — using a data-driven approach as you suggest — as adaptively identified high-bias regions in the data that would make the most sense to debias your predictions on.
>
> 2. Regarding avoiding explicit computation of best-response events for all agents: This is a great question, and answering it in specific settings is possible and has led to novel follow-up results, in particular addressing the desideratum you describe where we may wish for no swap regret for all downstream agents with utility functions in a certain class.
> For instance, the follow-up work of Roth and Shi (2024) give a simple method for doing this using our algorithm. Their observation is that for any (linear) utility function, best response regions are convex --- so it suffices to use our algorithm to produce unbiased predictions for all regions defined by the convex hull of points in our prediction space. Each of these defines an event that can be plugged into our algorithm. For one dimensional outcome spaces, these are simply intervals. This gives us a finite collection of events that includes the best response events for all possible downstream agents, even though we don’t have knowledge of their utility functions.
> The upshot of the described construction of Roth and Shi is that relative to implementing full calibration (which you point out in your review as another agent-agnostic approach), you get better regret rates — O(sqrt(T)) vs. O(T^2/3) in one dimension, and much better improvements in higher dimensions, since bounds for full online calibration scale exponentially with the dimension. Again, their bounds follow from simply instantiating our algorithm with the right choice of events.
> We look forward to further applications of our framework in this vein.
>
> 3. Regarding the notion of “sufficient statistics”:  We have referred to the state vectors’ entries as “sufficient statistics” in the intro, to informally convey their meaning. Rigorously, as you indeed say, we define them as the (finite as we require) collection of quantities that the agent’s utility mapping is a function of, for every action. Whether or not they represent some statistics of some implicit/underlying distributions on which the agents’ utilities depend, does not affect our framework. In this work, we focus on establishing the foundational case where the utility is  an affine function of these statistics. However, this linearity should not be viewed as some very restrictive assumption that would preclude the use of the framework for nonlinear utilities, but rather as the key tool for nonlinear extensions: this is similar to how linear optimization serves as the core of most convex/nonconvex optimization methods. To be more specific, lots of very important classes of (utility) functions have small functional bases (needed to keep our event collection small), and linearize over the states once an appropriate representation map is applied to the state vectors. E.g. for polynomials of fixed degree, for the states we would consider the moments up to the d-th, and this can be taken far beyond polynomial utilities, too, provided some other smoothness guarantees on the function class are given. For instance a very recent preprint (https://arxiv.org/abs/2502.12564) looks at such extensions and once again directly applies our framework and algorithm on those.

---

> > ### Comment · Reviewer_MMHe · 2025-04-04
> >
> > thanks for the detailed response to my comments. I'm happy with the current submission and would like to see at the conference, however some of these clarifications can go into the main text.

---

### Official Review · Reviewer_XKgH · 2025-03-20

**Overall Recommendation:** 4

**Summary:**

This paper introduces a general decision-making framework for the design of efficient algorithms with the objective of producing multi-dimensional forecasts in adversarial, sequential decision-making environments
The main ides is to first design a way to sequentially predict the future state of the environment, ensuring low bias across a polynomial number of conditioning events, and second to design regret-minimizing agents that essentially assume perfect knowledge of the next state.
Therefore, the framework shifts the traditional focus from direct regret minimization to predicting a sequence of "sufficient statistics" of the environment, as it suffices for the agents to best-respond with respect to the predicted states.
By guaranteeing small cumulative bias of the state predictions, the overall regret of the agents can be bounded in a nontrivial way while guaranteeing efficiency at the same time by cleverly designing the event collection of interest for the sequential decision-making problem at hand.
The authors demonstrate that these almost unbiased state predictions can be used to guarantee strong conditional regret bounds, even when dealing with complex online combinatorial optimization problems and multiple agents.
They also show that the algorithm obtained via this framework can efficiently achieve the $T^{2/3}$ rate in online multicalibration.

**Claims And Evidence:**

The theoretical claims are clearly stated with complete proofs.
The authors provide a detailed description of the proposed framework and the resulting algorithms, which is well-structured and easy to follow.

**Essential References Not Discussed:**

To the best of my knowledge, the paper seems to discuss the relevant references.

**Experimental Designs Or Analyses:**

N/A

**Methods And Evaluation Criteria:**

N/A

**Other Comments Or Suggestions:**

- Line 223, second column: shouldn't there be some expected value since $n\_T$ is defined as an expectation (Definition 2.2) whereas the realized predictions $\\hat s\_t$ are random variables?
- Line 247, second column: "(1)" seems to be superfluous.
- Line 260, second column: the sum $\\sum\_{p \\in [0,1]}$ is somewhat ill-defined at first glance because of the uncountable range. Then, given the term in the sum, it would be clearer (and equivalent) to have $p \\in \{p\_1, \\dots, p\_T\}$. Even just a comment would make this clear.
- Line 271, second column: the first comma should be a period.
- Lien 298, second column: $nK$ instead of $nd$.
- Line 368, first column: $O(nK)$ is actually just $nK$ to be precise.
- Line 384, second column: "for any set" instead of "for any of a set".

**Other Strengths And Weaknesses:**

I found the proposed methodology to be very interesting and the paper to be well-written.
The authors provide a clear and detailed description of the proposed framework and the resulting algorithms, which is well-structured and easy to follow.
I also appreciated the instantiation of the framework to specific problems, which not only demonstrates the versatility and effectiveness of the proposed approach by deriving improved results in relevant problems, but also helps the reader in further understanding how to apply the framework to other problems.

**Questions For Authors:**

Could you clarify the relevance of the current work with respect to any potential work that extended or improved the framework since the publication of its preprint version?

**Relation To Broader Scientific Literature:**

The authors provide a comprehensive overview of the related prior work and discuss their contribution with respect to existing results.
My only concern is that the paper, as the authors explicitly state, had an extended preprint version that appeared in 2023.
Even so, the authors are careful in pointing out that the proposed method lead to follow-up work that achieved improved results for specific sequential decision-making problems.
However, there is a chance that a more recent paper could potentially already contain improved results.
The authors should further clarify this specific point, for instance by explicitly stating that the proposed technique is not obsolete and currently remains a main reference.

**Theoretical Claims:**

Yes, I verified the proofs in this paper and they seem correct.

---

> ### Author Rebuttal · Authors · 2025-04-01
>
> We thank the reviewer for the careful reading and positive assessment of our paper, and appreciate the detailed comments. We will incorporate, in the updated version of the paper, the modifications proposed in Other Comments and Suggestions.
>
> As for the relevance of the preprint to future work that followed up on it, such as the cited references in the submission: Indeed, our proposed framework and the algorithm we develop within it is not obsolete and remains state of the art and continues to be used. Subsequent work has not generalized or subsumed our general framework. Rather, follow-up papers have used our framework and algorithm and its generic guarantees as a technical tool towards achieving goals in concrete settings (such as high-dimensional contract design, or no regret guarantees for infinite families of downstream agents, etc) — which is the type of follow-up work that we have envisioned for our paper from the beginning.
>
> To summarize, our framework remains state-of-the-art, and subsequent work has used it for various interesting applications.
>
> Please feel free to look at the responses to other reviewers for some brief points on a couple of follow-ups and on ways in which these follow-ups use our framework.

---

### Decision · Program_Chairs · 2025-05-01

**Decision:**

Accept (oral)

**Comment:**

This paper develops algorithms for predicting underlying “states” of a sequential process. These state predictions, if sufficiently accurate, can then be used by downstream algorithms to minimize stronger notions of regret such as swap regret. The state predictions are generated by a kind of online analog of moment matching. The reviewers all were impressed with the strong theoretical contributions in this paper.